# Reduced synchroneity of intra-islet Ca²⁺ oscillations in vivo in *Robo*-deficient β cells

Melissa T Adams[1], JaeAnn M Dwulet[2], Jennifer K Briggs[2], Christopher A Reissaus[3], Erli Jin[4], Joseph M Szulczewski[1], Melissa R Lyman[1], Sophia M Sdao[4], Vira Kravets[2,5], Sutichot D Nimkulrat[1], Suzanne M Ponik[1], Matthew J Merrins[4], Raghavendra G Mirmira[6], Amelia K Linnemann[3], Richard KP Benninger[2,5], Barak Blum[1]*

[1]Department of Cell and Regenerative Biology, University of Wisconsin-Madison, Madison, United States; [2]Department of Bioengineering, University of Colorado Denver, Anschutz Medical Campus, Aurora, United States; [3]Herman B Wells Center for Pediatric Research and Center for Diabetes and Metabolic Diseases, Indiana University School of Medicine, Indianapolis, United States; [4]Department of Medicine, Division of Endocrinology, Diabetes, and Metabolism, University of Wisconsin-Madison, Madison, United States; [5]Barbara Davis Center for Diabetes, University of Colorado Anschutz Medical Campus, Aurora, United States; [6]Kovler Diabetes Center and the Department of Medicine, University of Chicago, Chicago, United States

*For correspondence:
bblum4@wisc.edu

Competing interests: The authors declare that no competing interests exist.

**Abstract** The spatial architecture of the islets of Langerhans is hypothesized to facilitate synchronized insulin secretion among β cells, yet testing this in vivo in the intact pancreas is challenging. Robo βKO mice, in which the genes *Robo1* and *Robo2* are deleted selectively in β cells, provide a unique model of altered islet spatial architecture without loss of β cell differentiation or islet damage from diabetes. Combining Robo βKO mice with intravital microscopy, we show here that Robo βKO islets have reduced synchronized intra-islet Ca²⁺ oscillations among β cells in vivo. We provide evidence that this loss is not due to a β cell-intrinsic function of Robo, mis-expression or mis-localization of Cx36 gap junctions, or changes in islet vascularization or innervation, suggesting that the islet architecture itself is required for synchronized Ca²⁺ oscillations. These results have implications for understanding structure-function relationships in the islets during progression to diabetes as well as engineering islets from stem cells.

## Introduction

The islets of Langerhans, which comprise the endocrine pancreas, are highly organized microorgans responsible for maintaining blood glucose homeostasis. Islets are composed of five endocrine cell types (α, β, δ, PP, and ε), which, in adult rodents, are arranged such that the β cells cluster in the core of the islet, while other non-β endocrine cells populate the islet periphery (*Orci and Unger, 1975*; *Pfeifer et al., 2015*; *Erlandsen et al., 1976*). This configuration heavily prioritizes homotypic over heterotypic contacts between endocrine cells. Mature human islet architecture is more complex, and, though its exact organization pattern is still debated, it appears to follow a non-random distribution of the different endocrine cell types (*Bonner-Weir et al., 2015*; *Bosco et al., 2010*; *Brissova et al., 2005*; *Cabrera et al., 2006*; *Dybala and*

*Hara, 2019*) and prioritize homotypic over heterotypic interactions between endocrine cell types (*Hoang et al., 2014*; *Kilimnik et al., 2012*). Though islet architecture is thought to be fairly static in healthy adult humans and rodents, it is quite dynamic during development, pregnancy, and diabetes (*Dolenšek et al., 2015*; *Miller et al., 2009*; *Nair and Hebrok, 2015*; *Sharon et al., 2019*; *Sznurkowska et al., 2020*; *Bocian-Sobkowska et al., 1999*; *Kharouta et al., 2009*; *Rieck and Kaestner, 2010*). In diabetes, rodent and human islet architecture is disrupted, resulting in endocrine cell intermixing and a reduction in the ratios of homotypic versus heterotypic cell contacts (*Brereton et al., 2015*; *Kilimnik et al., 2011*; *Nir et al., 2007*; *Striegel et al., 2015*; *Xiao et al., 2018*). How islet architecture and endocrine cell-type sorting within the islet affect islet function remains poorly understood (*Steiner et al., 2010*).

Homotypic interactions between β cells are important for synchronous glucose-stimulated insulin secretion (GSIS). In GSIS, glucose from the blood enters β cells through glucose transporters and is metabolized, causing an increase in the ratio of intracellular ATP/ADP. This increase in ATP/ADP triggers closure of ATP-sensitive $K^+$ channels, resulting in membrane depolarization and the opening of voltage-gated $Ca^{2+}$ channels, triggering an influx of $Ca^{2+}$ into the cell, which in turn promotes exocytosis of insulin granules (*Ammälä et al., 1993*; *Ashcroft and Rorsman, 2013*; *Bertram et al., 2010*). This process is cyclical and thus oscillatory. In dispersed islets, which have no cell-cell contacts, β cells display heterogenous uncoordinated oscillations in membrane depolarization and $Ca^{2+}$ influx at both basal and elevated glucose, which results in high basal insulin secretion and low uncoordinated GSIS (*Halban et al., 1982*). In intact islets, where homotypic interactions between β cells are present, the oscillations in membrane potential, intracellular $Ca^{2+}$ ($[Ca^{2+}]_i$), and insulin secretion that underlie GSIS are respectively synchronous at elevated glucose and relatively silent at low glucose. This synchronous or 'pulsatile' pattern of insulin secretion from islets is thought to underlie pulsatile insulin levels in circulating blood, a quality important for keeping the peripheral tissues insulin sensitive, for robust liver response to insulin signaling, and for allowing time for replenishment of the readily releasable pool of insulin granules in β cells. Thus, perturbations in pulsatility are thought to contribute to diabetes pathology (*Lang et al., 1981*; *Matveyenko et al., 2012*; *Satin et al., 2015*; *Pedersen and Sherman, 2009*). Indeed, pulsatile insulin levels in circulating blood of mouse models of diabetes and patients with diabetes, pre-diabetes, and non-diabetic family members of diabetic patients are often disrupted (*Satin et al., 2015*; *O'Rahilly et al., 1988*).

One mechanism through which homotypic β cell-β cell contacts can synchronize GSIS is through electrical coupling. Within an islet, β cells are electrically coupled to their homotypic neighbors via varying levels of Connexin36 (Cx36) gap junctions that allow for exchange of cations between neighboring β cells (*Benninger et al., 2008*; *Farnsworth et al., 2014*). β cells coupled by these gap junctions display varying levels of excitability and metabolic rates, aspects which themselves are thought to display a non-random spatial organization (*Hraha et al., 2014*). This coupling of spatially organized heterogenous β cells populations creates an electrical syncytium that responds homogenously to glucose such that, at low glucose levels, insulin secretion is inhibited through repression of $[Ca^{2+}]_i$ oscillations across the islet, but at high glucose levels, pulsatile insulin secretion occurs through synchronous $[Ca^{2+}]_i$ oscillations that spread in fast waves across the islet from distinct initiation sites (*Benninger et al., 2008*; *Farnsworth et al., 2014*; *Hraha et al., 2014*; *Skelin Klemen et al., 2017*; *Benninger and Piston, 2014*; *Westacott et al., 2017a*; *Benninger et al., 2014*; *Kravets, 2020*; *Ravier et al., 2005*; *Speier et al., 2007*; *Head et al., 2012*). Indeed, in silico modeling experiments showed that decreasing the ratio of homotypic β cell-β cell nearest neighbors is predicted to result in perturbation to synchronous $[Ca^{2+}]_i$ oscillations (*Hoang et al., 2014*; *Head et al., 2012*; *Nittala et al., 2007*; *Hoang et al., 2016*). A highly functionally connected subpopulation of β cells, termed 'hub' or 'leader' cells, may also direct synchronous $[Ca^{2+}]_i$, oscillations by harboring a disproportionate amount of functional connections to other β cells, a property allowed for by a high frequency of homotypic interactions between β cells (*Johnston et al., 2016*; *Salem et al., 2019*).

It is thus hypothesized that disrupting proper endocrine cell type sorting in the islet in a way that distorts the relative amount of homotypic β cell-β cell contacts, even without affecting any other property of the cell, would be sufficient to disrupt synchronized oscillatory behavior among β cells. However, direct empirical evidence supporting this hypothesis is lacking. This is because of the fact that while most genetic mouse models that show abnormally disorganized islet architecture also display defects in glucose homeostasis (*Brereton et al., 2015*), the disrupted islet

architecture is usually linked to either developmental defects in β cell differentiation or maturation (*Hang et al., 2014*; *Yamagata et al., 2002*; *Gannon et al., 2000*; *Gu et al., 2010*; *Ahlgren et al., 1998*; *Borden et al., 2013*; *Doyle and Sussel, 2007*; *Sinagoga et al., 2017*; *Huang et al., 2018*; *Jimenez-Caliani et al., 2017*; *Bastidas-Ponce et al., 2017*; *Crawford et al., 2009*) or to pathologies related to β cell damage in diabetes (*Kilimnik et al., 2011*; *Nir et al., 2007*; *Szabat et al., 2016*; *Brissova et al., 2014*; *Baetens et al., 1978*; *Starich et al., 1991*). This introduces a strong confounding factor for studying the role of islet architecture in islet function. Therefore, it is difficult to disentangle the relative effect of β cell-intrinsic defects and whole-islet architectural defects, such as reduced ratio of homotypic β cell-β cell contacts, on perturbation of synchronous oscillations that underlie normal GSIS.

Recently, we have described a mouse model in which the cell-surface receptors *Robo1* and *Robo2* are deleted specifically in β cells (Robo βKO), resulting in disruption of canonical endocrine cell-type sorting within the islets (*Adams et al., 2018*). Unlike other models of disrupted islet architecture and endocrine cell-type sorting in the islet, the β cells in the islets of Robo βKO express normal levels of markers for β cell differentiation, functional maturity, and regulation of GSIS, and show normal β cell-intrinsic response to glucose. We reasoned that this model would allow us to test the role of islet architecture and endocrine cell-type sorting in regulating synchronous [Ca$^{2+}$]$_i$ oscillations in response to glucose among β cells in a fully differentiated, non-diabetic islet setting.

## Results

### Robo βKO islets have a decreased ratio of homotypic β cell-β cell contacts but their β cells are functionally mature

In silico simulations where the degree of β cell-β cell coupling is changed through a decrease in homotypic nearest neighbors predict that disruption of islet architecture will disrupt synchronous intra-islet [Ca$^{2+}$]$_i$ oscillations and pulsatile hormone secretion (*Hoang et al., 2014*; *Nittala et al., 2007*; *Hoang et al., 2016*). To test whether β cells in Robo βKO islets have a decreased frequency of homotypic β cell-β cell contacts compared to control islets, we performed nearest-neighbor analysis on islets from pancreatic sections of Robo βKO and control mice (*Figure 1A*). We found that Robo βKO islets possess a significantly decreased frequency of homotypic β cell-β cell contacts and a significantly increased frequency of heterotypic β cell contacts with either α or δ cells (*Figure 1B, C*). α and δ cells trended towards a higher ratio of homotypic contacts in Robo βKO compared to controls though this did not reach statistical significance (p=0.06) and showed no difference in the level of heterotypic contacts with each other compared to controls (*Figure 1D, E*). Thus, Robo βKO islets have a lower frequency of homotypic β cell-β cell contacts and increased frequency of β cell heterotypic contacts compared to control islets while non-β endocrine cells are less affected.

We have previously shown that genetic deletion of *Robo1* and *Robo2* selectively in β cells using either *Ins1$^{Cre}$;Robo1$^{Δ/Δ}$Robo2$^{flx/flx}$* or *Tg(Ucn3-Cre);Robo1$^{Δ/Δ}$Robo2$^{flx/flx}$* mice (Robo βKO) results in disrupted islet architecture and endocrine cell-type sorting without affecting β cell death or the expression of the β cell maturation markers MafA and Ucn3 (*Adams et al., 2018*). To verify that β cells in Robo βKO islet are more broadly mature, we expanded the analysis to look at transcript levels of 15 additional β cell maturity markers. RNA sequencing and differential gene expression analysis on FACS-purified β cells from both Robo βKO and control islets revealed no change in transcript levels of any hallmark β cell maturity or differentiation genes (*Figure 1—figure supplement 1*, *Supplementary file 1*). Thus, unlike other mouse models with disrupted islet architecture, β cells in Robo βKO islets appear to maintain maturity and differentiation despite loss of normal islet architecture.

Though Robo βKO β cells show normal maturity and differentiation as determined by RNA sequencing and immunostaining for maturity and differentiation factors, experiments have shown that the Slit-Robo signaling pathway is involved in the stimulus secretion cascade linking glucose to insulin secretion in in vitro cultured β cells (*Yang et al., 2013*). Confoundingly, our RNAseq analysis showed no change in the expression of the major genes responsible for regulation of Ca$^{2+}$ dynamics or GSIS between control and Robo βKO β cells (*Figure 1—figure supplement 1*). Thus to test whether Robo βKO β cells are able to undergo normal [Ca$^{2+}$]$_i$ oscillations in

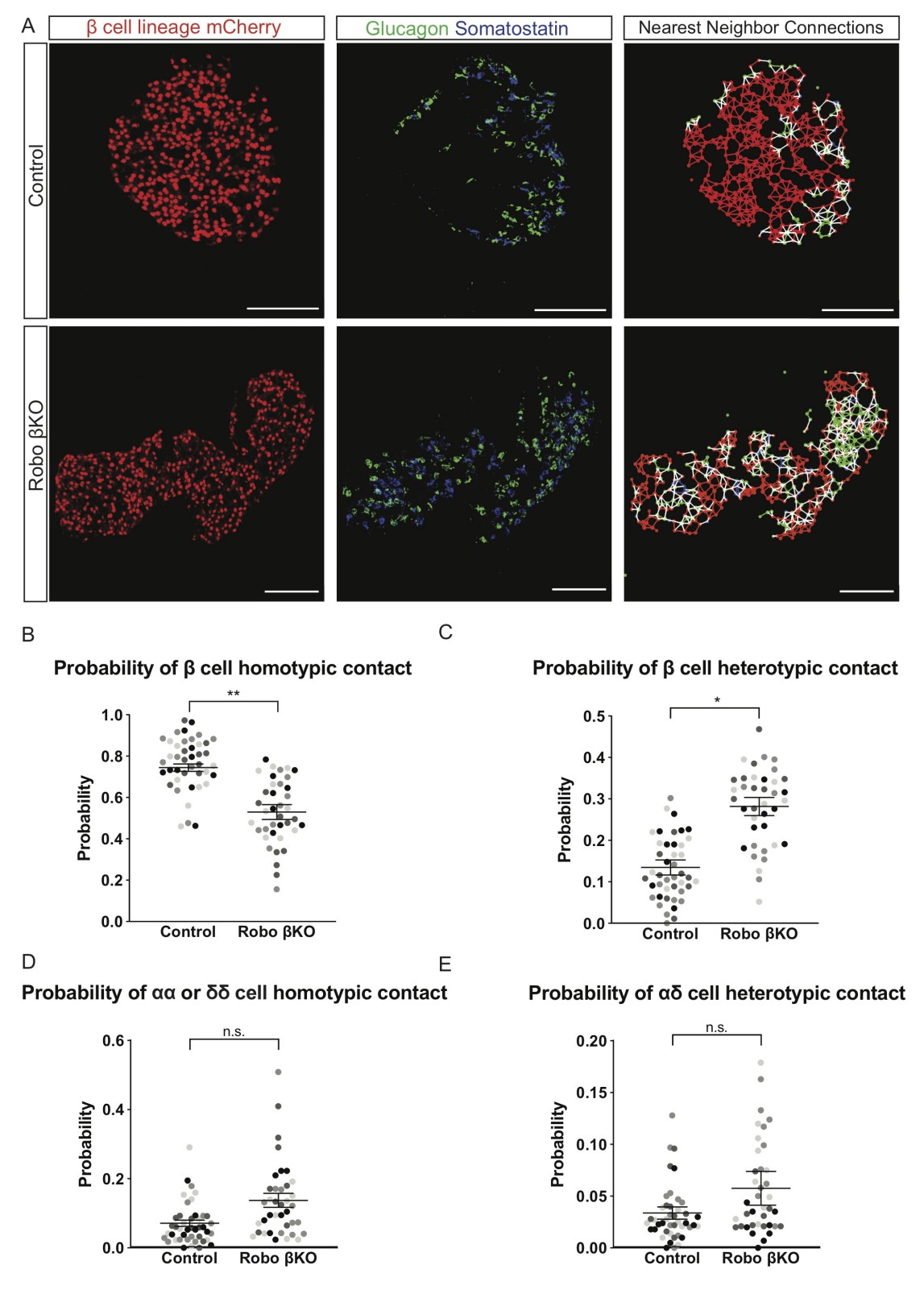

**Figure 1.** Robo βKO islets have a decreased ratio of homotypic nearest neighbors than controls. (**A**) Immunofluorescence images (left and middle panels) and cell connectivity maps generated by nearest-neighbor analysis (right panels) of control and Robo βKO islets. β cells (red), α cells (green), and δ cells (blue) are denoted by nodes on the connectivity maps. A line the same color as both nodes it connects denotes a homotypic interaction of that corresponding cell type. A white line connecting two nodes denotes a heterotypic interaction between cell types. Scale bars are 100 μm. (**B**)
*Figure 1 continued on next page*

Figure 1 continued

Probability of β cell homotypic contact in Robo βKO islets vs. controls. n = 4 mice; control 0.74 ± 0.02 SEM, Robo βKO 0.53 ± 0.04 SEM, p<0.005 t-test. (C) Probability of β cell heterotypic contacts in Robo βKO islets vs. controls, n = 4 mice, control 0.13 ± 0.02 SEM, Robo βKO 0.28 ± 0.02 SEM, p<0.05 MW. (D) Probability of αα or δδ homotypic contacts in Robo βKO islets vs. controls, n = 4 mice; control 0.07 ± 0.01 S.EM, Robo βKO 0.13 ± 0.02 SEM, p=0.06 MW. (E) Probability of α-δ heterotypic contact in Robo βKO islets vs. controls, n = 4 mice; control 0.03% ± 0.01 SEM, Robo βKO 0.06% ± 0.02 SEM, p=0.22 t-test. (B–E) Similar shaded points in graphs indicate islets from the same mouse, while mean and error bars represent statistics performed on average values from each mouse. Error bars show SEM. 9–11 islets from an individual mouse were measured as technical replicates, and the average values per mouse were used as biological replicates. MW: Mann–Whitney; SEM: standard error of the mean.

The online version of this article includes the following source data and figure supplement(s) for figure 1:

**Figure supplement 1.** Robo βKO islets retain β cell differentiation and maturity, and $Ca^{2+}$ regulatory genes.

**Figure supplement 1—source data 1.** RNA sequensing source data.

response to stimuli, we performed in vitro $[Ca^{2+}]_i$ imaging on single β cells from dissociated Robo βKO and control islets exposed to glucose followed by KCl (*Figure 2*). We found no difference in the proportion of β cells that undergo $[Ca^{2+}]_i$ oscillations in response to 10 mM glucose between control and Robo βKO β cells (*Figure 2A–C*). We also saw no significant difference in area under the curve (AUC) of $[Ca^{2+}]_i$ in response to 10 mM, in peak height corresponding to the first phase of insulin secretion, or in AUC $[Ca^{2+}]_i$ in response to KCl in control and Robo βKO β cells (*Figure 2D–H*). Together, this demonstrates that Robo βKO β cells are not defective in their ability to undergo $[Ca^{2+}]_i$ oscillations in response to glucose, suggesting that β cell-intrinsic factors mediated by *Robo* deletions do not have a strong impact on the $[Ca^{2+}]_i$ dynamics that underlie GSIS in individual β cells.

We reasoned that the altered degree of homotypic β cell-β cell interactions in Robo βKO islets, together with the retained β cell maturity and intact intrinsic β cell $[Ca^{2+}]_i$ dynamic functionality, provides a unique model by which to empirically test the hypothesis that endocrine cell-type organization within the islet affects synchronous oscillatory behavior in intra-islet β cells.

## Robo βKO islets display unsynchronized $Ca^{2+}$ oscillations in vivo

We thus set out to investigate how the reduced homotypic β cell-β cell interactions in Robo βKO islets affect synchronous β cell behavior by measuring dynamic $[Ca^{2+}]_i$ oscillations in response to glucose. Robo βKO islets are fragile in isolation and culture (*Adams et al., 2018*), making them unsuitable for in vitro analyses of whole-islet $[Ca^{2+}]_i$ oscillations. To overcome this limitation, we adopted an intravital $[Ca^{2+}]_i$ imaging method that enables imaging of islet $[Ca^{2+}]_i$ dynamics in situ within the intact pancreas (*Reissaus et al., 2019*). In brief, this method employs an intravital microscopy (IVM) platform and adeno-associated viral (AAV) delivery of insulin promoter-driven GCaMP6s, a fluorescent $Ca^{2+}$ biosensor, to quantitate β cell $[Ca^{2+}]_i$ dynamics in vivo in both Robo βKO and control islets. This method also allows for retention of the islet's in vivo microenvironment, blood flow, and innervation, thus providing a more realistic condition than in vitro approaches to study islet function allow for.

We verified that synchronous $[Ca^{2+}]_i$ oscillations are maintained in vivo in islets by measuring GCaMP6s intensity in β cells within AAV8-RIP-GCaMP6-infected islets of control (*Robo WT*) mice (*Figure 3*). As expected, control mice displayed whole-islet synchronous $[Ca^{2+}]_i$ oscillations when imaged at anywhere from ~0.03 to ~1 Hz for at least 10 min after glucose elevation (*Figure 3* and *Videos 1* and *2*). We quantified the level of synchroneity in intra-islet oscillations by analyzing the correlation between GCaMP6s active areas within individual islets (*Figure 3C, D*). In brief, this analysis measures the proportion of GCaMP6s active area within an islet where the normalized GCaMP6s intensity over time has a >0.7 Pearson's correlation coefficient. This is demonstrated by synchronous region maps, which show highly correlated areas of the islet in the same color, revealing that almost all of the active $[Ca^{2+}]_i$ area is synchronized within control islets (*Figure 3D*). While oscillations vary in frequency between islets, the area of highly correlated $[Ca^{2+}]_i$ oscillations between β cells within any one islet is very high, confirming that control islets possess highly synchronous intra-islet $[Ca^{2+}]_i$ oscillation in response to glucose in vivo (*Figure 3* and *Figure 4E*).

Conversely, we found that Robo βKO islets on average display less synchronous intra-islet $[Ca^{2+}]_i$ oscillations in vivo when imaged at speeds from ~0.03 Hz to ~1 Hz (*Figure 4*, *Figure 4—figure supplement 1*, and *Videos 3*, *4,* and *5*). Quantification of this asynchronous behavior through

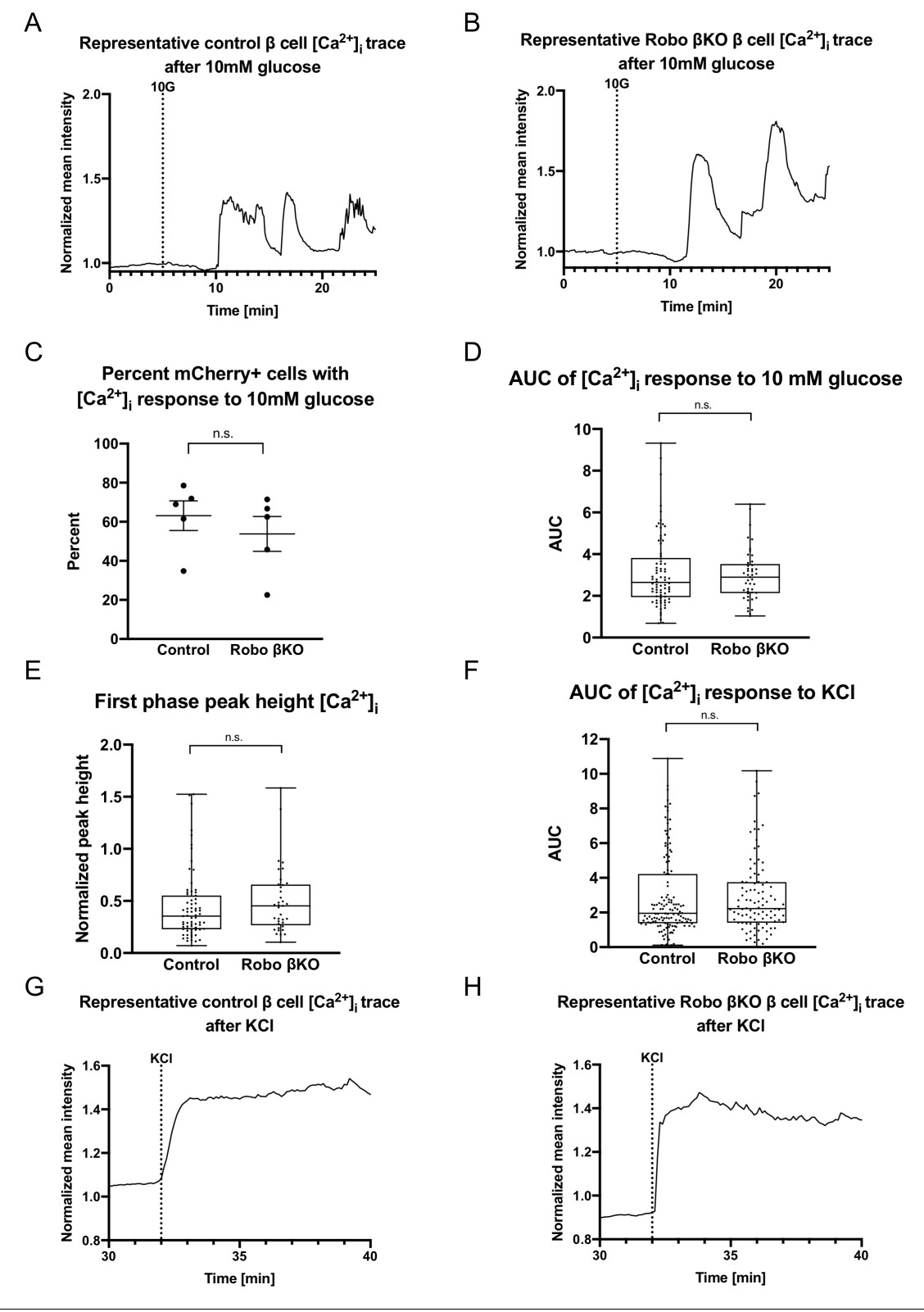

**Figure 2.** Dissociated Robo βKO β cells show no difference in glucose-stimulated $Ca^{2+}$ oscillations. (**A**) Representative $[Ca^{2+}]_i$ trace (Fura2) of a dispersed β cell from a control islet. 10G line marks the addition of 10 mM glucose. (**B**) Representative $[Ca^{2+}]_i$ trace (Fura2) of a dispersed β cell from a Robo βKO islet. 10G line marks the addition of 10 mM glucose. (**C**) Proportion of $[Ca^{2+}]_i$-responsive β cells in Robo βKO compared to controls; n = 5 mice from each genotype, control 63.15 ± 7.6%, Robo βKO 53.80 ± 8.9% SEM, p=0.45 t-test. Error bars shown are SEM. (**D**) Area under the curve (AUC)

*Figure 2 continued on next page*

*Figure 2 continued*

of $[Ca^{2+}]_i$ (Fura2) from control and Robo βKO single dispersed β cells in response to 10 mM glucose. n = 78 β cells from five mice for control and n = 45 β cells from five mice for Robo βKO, control 3.1 ± 0.20 SEM, Robo βKO 3.0 ± 0.18 SEM, p=0.87 MW. (E) Peak height of $[Ca^{2+}]_i$ corresponding to first phase insulin secretion from control and Robo βKO single dispersed β cells in response to 10 mM glucose. n = 71 β cells from five mice for control and n = 37 β cells from five mice for Robo βKO, control 0.44 ± 0.04 SEM, Robo βKO 0.50 ± 0.05 SEM, p=0.17 MW. (F) AUC of $[Ca^{2+}]_i$ (Fura2) from control and Robo βKO single dispersed β cells in response to KCl. n = 132 β cells from five mice for controls and n = 98 β cells from five mice for Robo βKO, control 2.9 ± 0.2 SEM, Robo βKO 2.9 ± 0.2 SEM, p=0.65 MW. (G) Representative $[Ca^{2+}]_i$ trace (Fura2) of a single dispersed β cell from a control islet. Line marks the addition of KCl. (H) Representative $[Ca^{2+}]_i$ trace (Fura2) of a dispersed β cell from a Robo βKO islet. Line marks the addition of KCl. MW: Mann–Whitney; SEM: standard error of the mean.

correlation analysis of GCaMP6s activity within individual Robo βKO islets revealed significant reduction in areas of intra-islet correlated oscillations compared to controls that could not be attributed to differences in the proportion of GCaMP6s-positive cells with elevated $[Ca^{2+}]_i$ activity between the two groups (*Figure 4E, F*, *Figure 4—figure supplement 2A*). Additionally, this difference in synchroneity could not be attributed to islet size as there was no correlation between size of synchronous area and islet size in either control or Robo βKO islets (*Figure 4—figure supplement 2D, E*). Further, some asynchronous Robo βKO islets showed spatially distinct areas that oscillated synchronously with immediate β cell neighbors but not with more distant regions of the islet (*Figure 4C, D*, *Figure 4—figure supplement 1*, and *Videos 3, 4,* and *5*). However, a subset of Robo βKO islets imaged showed synchronous $[Ca^{2+}]_i$ activity in greater than 80% of GCaMP6s-positive areas (*Figure 4E*, *Figure 4—figure supplement 3*, and *Videos 6* and *7*), which is similar to levels in control islets. This synchronous population showed no difference in phase lag or wave speed when compared to controls, indicating that they share similar wave dynamics in addition to synchroneity with control islets (*Figure 4—figure supplement 2B, C*). The existence of this highly synchronous population of Robo βKO islets further suggests that the ability of individual β cells to oscillate $[Ca^{2+}]_i$ in response to stimuli is unaffected by deletion of *Robo*. This supports the idea that Robo βKO β cells do not have intrinsic defects in $[Ca^{2+}]_i$ dynamics in response to glucose, and that instead these oscillation defects are due to β cell-extrinsic factors within the islets of Robo βKO mice.

## Robo βKO islets have altered functional network properties

To gain a more in-depth understanding of how $Ca^{2+}$ dynamics are altered in Robo βKO islets in vivo, we performed networks analysis based on the methods previously described by *Stožer et al., 2013*. We thus analyzed how oscillations in β cells within an islet correlate with each other and created a network map of single planes from $[Ca^{2+}]_i$ recordings that had high enough resolution to identify individual cells. In these network maps, the correlation threshold for functional connections was set to R = 0.95, thus any β cells that had a correlation coefficient R $\geq$ 0.95 were considered functionally connected (*Figure 5A*). In line with results from the correlated area analysis (*Figure 4E*), the average functional connectivity between all β cell pairs within an islet was significantly lower in Robo βKO islets compared to controls, with Robo βKO islets that showed high areas of correlation also showing network connectivity levels similar to controls (*Figure 5A, B*). This decrease in functional connectivity was also demonstrated by the difference in the probability distributions of average percent links in control and Robo βKO islets (*Figure 5C*). Interestingly, we also found that the standard deviation of average islet connectivity was significantly lower in Robo βKO islets compared to controls, indicating that individual β cells within Robo βKO islets showed a larger variation in connectedness than β cells in control islets (*Figure 5D*). We also measured the number of highly connected β cells within the islets, or 'hub' cells, by measuring the percent of β cells that were functionally connected to $\geq$25% of the islet. We observed a trend towards a lower percentage of hub cells per islet in Robo βKO compared to controls, though this did not reach the threshold for statistical significance likely because of an outlier in the Robo βKO group (Mann–Whitney [MW] p=0.07). When we performed outlier analysis on these data, it revealed a significant decrease in the percent of hub cells in Robo βKO vs. control islets (MW p<0.05*; Figure 5E*).

Further, we measured the average clustering coefficient and global efficiency of these networks in order to characterize their 'small world' properties which have been shown to characterize wild-type islet $[Ca^{2+}]_i$ oscillations networks (*Stožer et al., 2013*; *Watts and Strogatz, 1998*). Small world networks are characterized by a high level of local clustering that is reflected by a high average

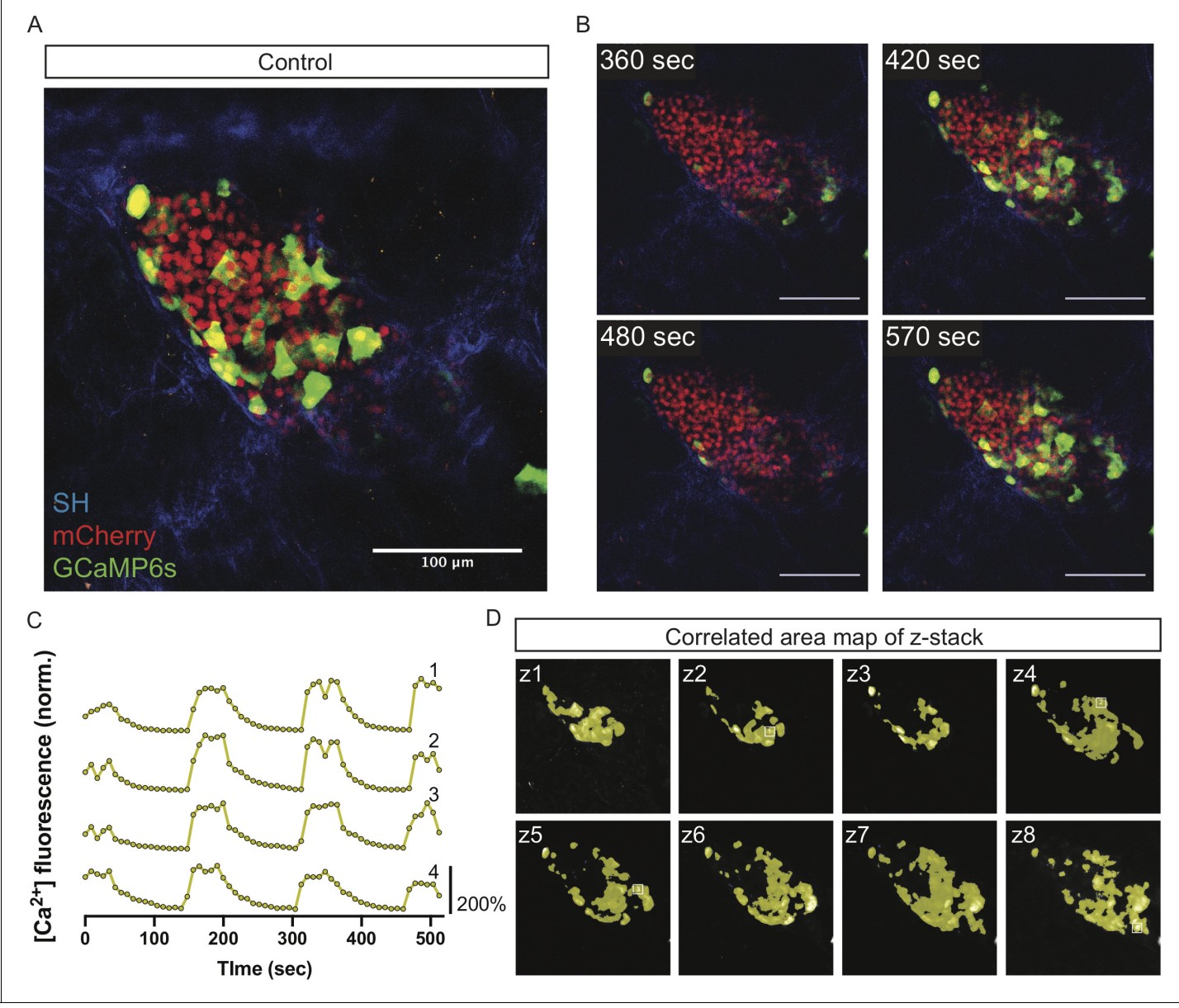

**Figure 3.** Control islets show highly synchronized whole-islet [Ca$^{2+}$]$_i$ oscillations. (**A**) Control islet in vivo in an *AAV8-RIP-GCaMP6s*-injected mouse showing GCaMP6s in green, nuclear mCherry β cell lineage tracing in red, and collagen (second harmonic) in blue. (**B**) Stills from a recording over one oscillation period from control islet in *Video 1*, when blood glucose level was >200 mg/dL following IP glucose injection. Video was recorded for 10 min with an acquisition speed of ~0.1 Hz. (**C**) Representative time courses of [Ca$^{2+}$]$_i$ activity in four individual areas from control islet in *Video 1* showing high correlation in activity over 97.2% of the active islet area. Time courses are normalized to average fluorescence of individual area over time. Similar color indicates that the time courses have a Pearson's correlation coefficient of ≥0.70 and matches the region of coordination that is seen in (**D**). (**D**) False color map of top four largest coordinated areas across z-stack of control islet from analysis in (**C**). Areas used in time courses in (**C**) are labeled. Scale bars are 100 μm.

clustering coefficient (C$_{avg}$), but also a high level of integration that is reflected by a high global efficiency (E$_{glob}$). In our analysis, we saw a trend toward decrease in both C$_{avg}$ and E$_{glob}$ in Robo βKO islets compared to controls though this did not quite reach statistical significance (p=0.05 for both MW; *Figure 4—figure supplement 2F, G*), indicating that Robo βKO islets act less like small world networks than control islets.

Altogether, network analysis suggests that Robo βKO islets have higher variability in functional connections per β cell within an islet, are less functionally connected on average, have a lower

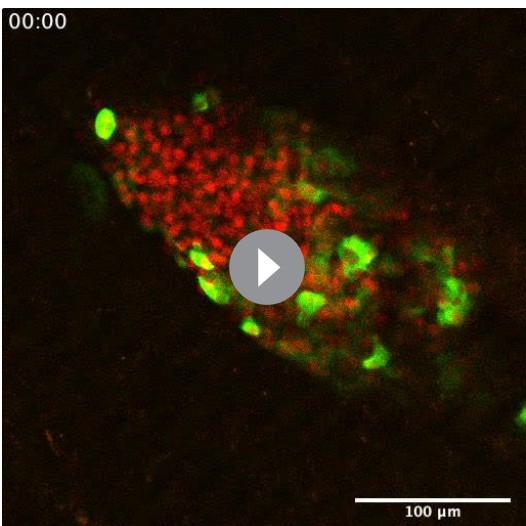

**Video 1.** Control islets show highly synchronized $[Ca^{2+}]_i$ oscillations. Intravital time-course video of an islet within the in vivo pancreas of a control β cell lineage-traced mouse infected with *AAV8-Ins1-GcaMP6s*. Lineage-traced β cells are marked by mCherry in red and GcaMP6s in green. Mouse was injected IP with glucose, and video was recorded once blood glucose levels reached >200 mg/dL. Z-stacks of eight slices each 8 µm apart were recorded at ~0.1 Hz over 10 min. Scale bar is 100 µm. Time stamp shown in the upper-left corner shows time of image in min:s. https://elifesciences.org/articles/61308#video1

**Video 2.** Control islets show highly synchronized $[Ca^{2+}]_i$ oscillations. Intravital time-course video of an islet within the in vivo pancreas of a control β cell lineage-traced mouse infected with *AAV8-ins1-GCaMP6s*. Lineage-traced β cells are marked by mCherry in red and GCaMP6s in green. Mouse was injected IP with glucose, and video was recorded once blood glucose levels reached >200 mg/dL. Z-stacks of three slices each 8 µm apart were recorded at ~0.2 Hz over 10 min. Scale bar is 100 µm. Time stamp shown in the upper-left corner shows time of image in min:s. https://elifesciences.org/articles/61308#video2

frequency of hub cells, and are less small world-like when compared to controls. This supports a scenario in which having less homotypic β cell-β cell interactions reduces the level of coupling within an islet.

## Robo βKO islets do not show defects in innervation and vascularization

One possible explanation for the observed disruption of intra-islet $[Ca^{2+}]_i$ oscillations in Robo βKO islets is through disruptions in innervation or vasculature within the islet as these factors have been implicated in controlling the ability of β cells to synchronize and the level of glucose exposure across the islet (*Eberhard and Lammert, 2009*). To assess whether the amount of innervation within the islets of Robo βKO differs from that of control islets, we stained pancreatic sections for the pan-neuronal marker Tubb3 and quantified the area of nerves normalized to islet area, and saw no difference in innervation between control and Robo βKO islets (*Figure 6—figure supplement 1*). To determine if there were changes in the amount of intra-islet vasculature in Robo βKO mice, we quantified the amount of matrix components secreted by vessels as a surrogate for vasculature (laminin and collagen IV) in Robo βKO and control islets. We observed no significant difference in area of vessel matrix components between Robo βKO and control islets for either of these proteins (*Figure 6A–D*). Despite the similar amount of vasculature in Robo βKO islets, it is possible that the pattern of vascularization may differ in such a way that delivery of glucose to certain regions of the islet is perturbed or delayed due to insufficient vascular coverage throughout the islet. To test whether glucose perfusion from the blood vessels was similar across the islet in Robo βKO compared to controls, we performed intravital imaging of islets during intravenous injection of the fluorescent glucose analog 2-NBDG (*Figure 6E–K* and *Videos 8* and *9*). We then assessed the timing of glucose arrival to areas <10 µm and >10 µm from the closest vessel to assess whether glucose can reach all portions of Robo βKO islets within a similar time frame as controls (*Figure 6F–H* and *Videos 8* and *9*). No

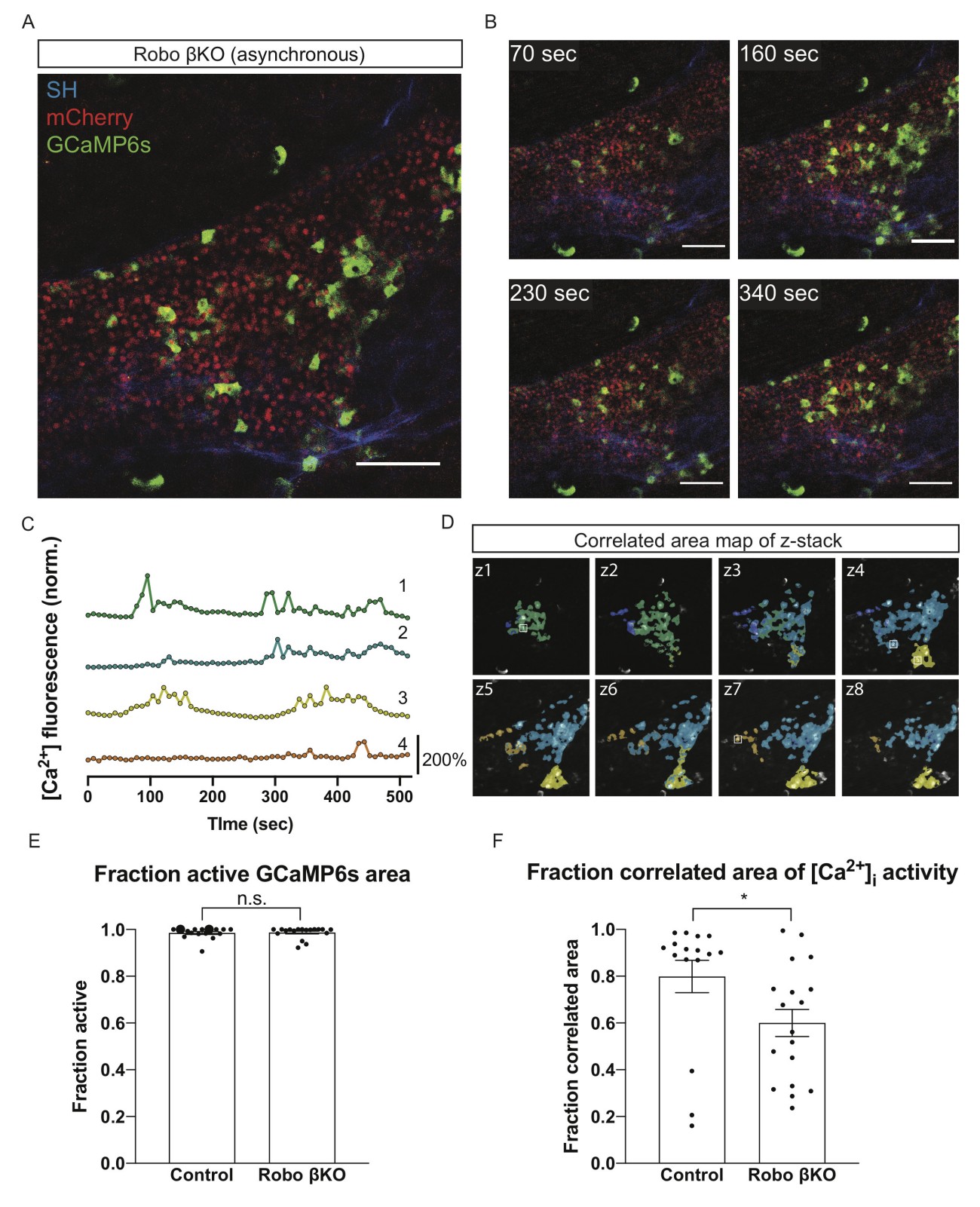

**Figure 4.** Robo βKO islets show decreased synchronization of whole-islet [Ca²⁺]ᵢ oscillations. (**A**) Robo βKO islet in vivo in an *AAV8-RIP-GCaMP6s-* injected mouse showing GCaMP6s in green, nuclear mCherry β cell lineage tracing in red, and collagen (second harmonic) in blue. (**B**) Stills from a recording over one oscillation period from Robo βKO islet in *Video 4*, when blood glucose level was >200 mg/dL after IP glucose injection. Video was recorded for 10 min with an acquisition speed of ~0.1 Hz. (**C**) Representative time courses of [Ca²⁺]ᵢ activity in five individual areas from Robo βKO islet

*Figure 4 continued on next page*

Figure 4 continued

in *Video 4*, showing high correlation in activity over 56.1% of the active islet area. Time courses are normalized to average fluorescence of individual area over time. Similar color indicates that the time courses have a Pearson's correlation coefficient of $\geq 0.70$ and matches the region of coordination that is seen in (D). (D) False color map of top four largest coordinated areas across z-stack of Robo βKO islet from analysis in (C). Areas used In (C) for traces are labeled. (E) Fraction of active islet area showing elevated $[Ca^{2+}]_i$ activity for control and Robo βKO islets. Control n = 16 islets collectively from seven mice, Robo βKO n = 18 islets collectively from nine mice, control 0.99 ± 0.006 SEM, Robo βKO 0.99 ± 0.006 SEM, p = 0.93 MW. (F) Largest fraction of area in islet exhibiting coordinated $[Ca^{2+}]_i$ oscillations for control and Robo βKO islets. Control n = 16 islets collectively from seven mice, Robo βKO n = 18 islets collectively from nine mice, control 0.80 ± 0.07 SEM, Robo βKO 0.60 ± 0.06 SEM, p<0.05 MW. Each islet was treated as a biological replicate for intravital oscillation experiments. Oscillation data was collected from five separate experiments. Error bars shown are SEM. Scale bars are 100 µm. MW: Mann–Whitney; SEM: standard error of the mean.

The online version of this article includes the following figure supplement(s) for figure 4:

**Figure supplement 1.** Robo βKO islets show uncoordinated whole-islet $[Ca^{2+}]_i$ oscillations.

**Figure supplement 2.** Robo βKO islets show uncoordinated whole-islet $[Ca^{2+}]_i$ oscillations regardless of imaging speed or islet size while a subset display wildtype-like wave properties.

**Figure supplement 3.** A subset of Robo βKO islets show coordinated whole-islet $[Ca^{2+}]_i$ oscillations.

---

significant differences in timing of glucose arrival between randomly sampled regions of the islet adjacent to blood vessels and regions >10 µm from the closest vessel in control and Robo βKO were seen (*Figure 6F, G*). We also saw no change in the AUC of average normalized 2-NBDG intensity in regions close to vessels within the same islet in controls compared to Robo βKO and in regions far from vessels between controls and Robo βKO (*Figure 6H, I*). Further, to measure the variability of 2-NBDG perfusion across individual islets, we measured the standard deviation of areas <10 µm from vessels within the same islet and areas >10 µm within the same islets in control and Robo βKO and saw no difference (*Figure 6J, K*). Altogether, this indicates that all areas of the islet are exposed to glucose at essentially the same time in both control and Robo βKO islets.

## Robo βKO islets retain normal expression levels and localization of Cx36 gap junctions

Another possible explanation for the loss of synchronized whole-islet $[Ca^{2+}]_i$ oscillations in Robo βKO is that the gap junction protein Cx36 is mis-localized or mis-expressed when *Robo* is deleted, and thus cell coupling is inhibited. Indeed, the phenotype described above for Robo βKO islets is reminiscent of that observed in mice heterozygous for a *Gjd2*-null allele (*Benninger et al., 2008*; *Ravier et al., 2005*). To test whether Cx36 is mis-localized in Robo βKO β cells, we stained for Cx36 along with F-actin to visualize the cell borders in β cell lineage-traced Robo βKO and control tissue sections (*Figure 7A*) and saw that in Robo βKO islets Cx36 still localized normally to the β cell borders. Further, to test whether Robo βKO mice downregulate Cx36 in islets, we measured the area of Cx36 protein immunofluorescence normalized to islet area in Robo βKO and control islets (*Figure 7B, C*). We found no difference in Cx36 area between Robo βKO islets and controls (*Figure 7C*). We also verified the specificity of our Cx36 antibody by staining $Cx36^{-/-}$ tissue

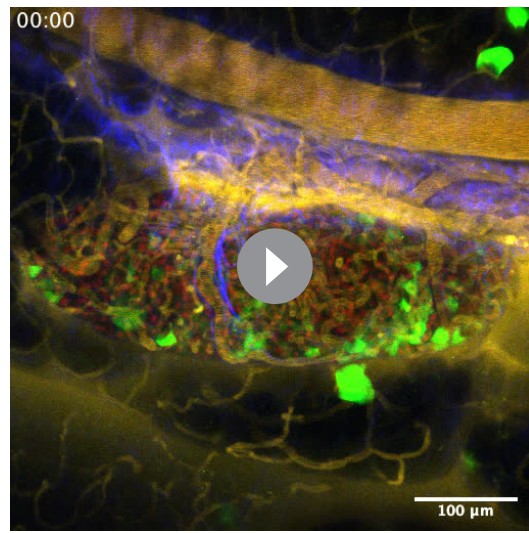

**Video 3.** Robo βKO islets show unsynchronized $[Ca^{2+}]_i$ oscillations. Intravital time-course video of an islet within the in vivo pancreas of a Robo βKO β cell lineage-traced mouse infected with *AAV8-ins1-GCaMP6s* and retroorbitally injected with rhodamine-dextran to mark vasculature. Lineage-traced β cells are marked by mCherry in red and GCaMP6s in green, and vasculature is shown in yellow. Mouse was injected IP with glucose, and video was recorded once blood glucose levels reached >200 mg/dL. Z-stacks of 12 slices each 8 µm apart were recorded at ~0.03 Hz over 10 min. Scale bar is 100 µm. Time stamp shown in the upper-left corner shows time of image in min:s. https://elifesciences.org/articles/61308#video3

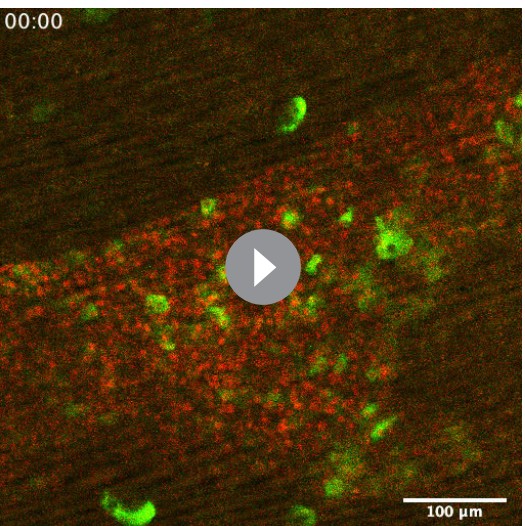

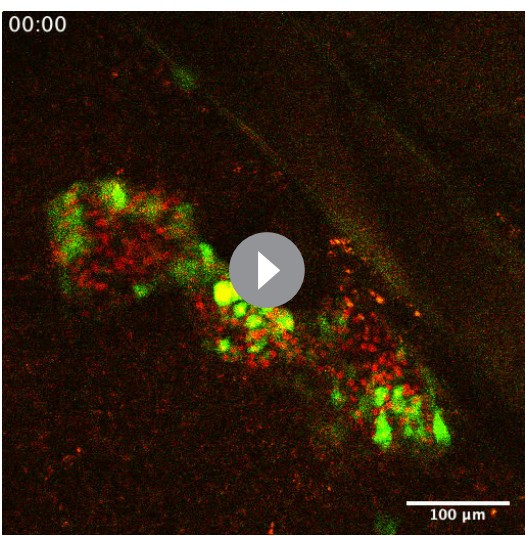

**Video 4.** Robo βKO islets show unsynchronized [Ca$^{2+}$]$_i$ oscillations. Intravital time-course video of an islet within the in vivo pancreas of a Robo βKO β cell lineage-traced mouse infected with *AAV8-ins1-GCaMP6s*. Lineage-traced β cells are marked by mCherry in red and GCaMP6s in green. Mouse was injected IP with glucose, and video was recorded once blood glucose levels reached >200 mg/dL. Z-stacks of eight slices each 8 μm apart were recorded at ~0.1 Hz over 10 min. Scale bar is 100 μm. Time stamp shown in the upper-left corner shows time of image in min:s.
https://elifesciences.org/articles/61308#video4

**Video 5.** Robo βKO islets show unsynchronized [Ca$^{2+}$]$_i$ oscillations. Intravital time-course video of an islet within the in vivo pancreas of a Robo βKO β cell lineage-traced mouse infected with *AAV8-ins1-GCaMP6s*. Lineage-traced β cells are marked by mCherry in red and GCaMP6s in green. Mouse was injected IP with glucose, and video was recorded once blood glucose levels reached >200 mg/dL. Z-stacks of three slices each 8 μm apart were recorded at ~0.2 Hz over 10 min. Scale bar is 100 μm. Time stamp shown in the upper-left corner shows time of image in min:s.
https://elifesciences.org/articles/61308#video5

sections, which showed no staining within the islet (*Figure 7B*). This is consistent with our RNAseq analysis that showed no change in Cx36 expression in Robo βKO compared to controls (*Figure 1— figure supplement 1*). Overall, this suggests that loss of synchronous intra-islet [Ca$^{2+}$]$_i$ oscillations is not due to decreased expression or mis-localization of Cx36 in Robo βKO β cells. Instead, this is consistent with a scenario in which coupling within Robo βKO islets is decreased due to a decrease in the ratio of β-β cell contacts rather than a loss of the gap junction machinery itself.

## Discussion

In this study, we provide evidence for the importance of islet architecture for proper islet function in vivo. We show that in Robo βKO islets, which possess disorganized islets architecture that results in a lower frequency of homotypic β cell-β cell interaction, synchronized [Ca$^{2+}$]$_i$ oscillations in the islet are disrupted. This disruption of synchronized [Ca$^{2+}$]$_i$ oscillations is not due to loss of functional β cell maturation and identity, altered Robo-mediated β cell-intrinsic defects in glucose-stimulated Ca$^{2+}$ dynamics, loss of Cx36 expression or localization to the β cell membrane, loss of innervation, or change in amount or pattern of islet vascularization, suggesting that endocrine cell-type sorting within the islet is, by itself, important for islet function, at least in the aspect of coordinated [Ca$^{2+}$]$_i$ oscillations among β cells.

Robo, and its ligand Slit, have been previously shown to affect [Ca$^{2+}$]$_i$ oscillations in β cells in vitro (*Yang et al., 2013*). However, Robo-mediated β cell-intrinsic effects are likely not the cause of asynchronous in vivo [Ca$^{2+}$]$_i$ oscillations in Robo βKO islets. This is supported by two independent observations: (1) single-cell [Ca$^{2+}$]$_i$ oscillations in dissociated β cells triggered by glucose stimulus in vitro are similar between Robo βKO and control, and (2) a subset of Robo βKO islets analyzed in vivo still show synchronized whole-islet [Ca$^{2+}$]$_i$ oscillations despite the absence of Robo. Further, this phenotypic heterogeneity is likely not due to an incomplete deletion of *Robo* because during Ca$^{2+}$ imaging experiments all islets were detected by the fluorescent labeling of β cells with the H2B-mCherry

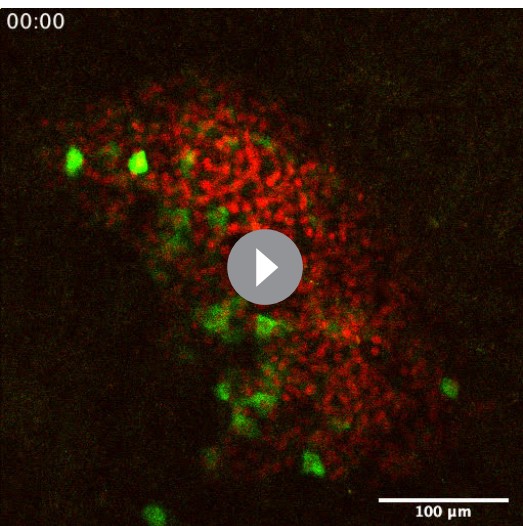

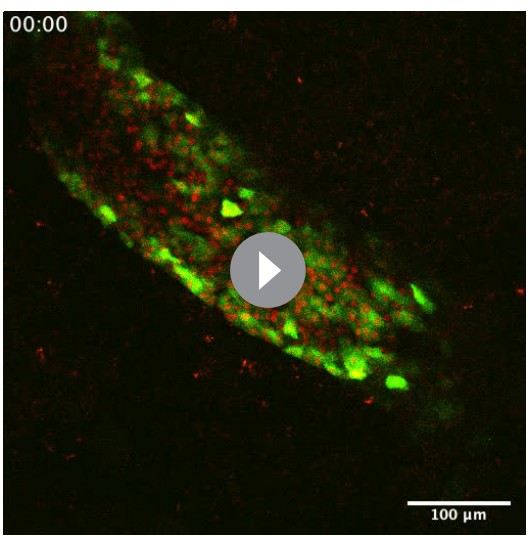

**Video 6.** A subset of Robo βKO islets retain synchronized $[Ca^{2+}]_i$ oscillations. Intravital time-course video of an islet within the in vivo pancreas of a Robo βKO β cell lineage-traced mouse infected with AAV8-ins1-GCaMP6s. Lineage-traced β cells are marked by mCherry in red and GCaMP6s in green. Mouse was injected IP with glucose, and video was recorded once blood glucose levels reached >200 mg/dL. Z-stacks of eight slices each 8 μm apart were recorded at ~0.1 Hz over 10 min. Scale bar is 100 μm. Time stamp shown in the upper-left corner shows time of image in min:s.
https://elifesciences.org/articles/61308#video6

**Video 7.** A subset of Robo βKO islets retain synchronized $[Ca^{2+}]_i$ oscillations. Intravital time-course video of an islet within the in vivo pancreas of a Robo βKO β cell lineage-traced mouse infected with AAV8-ins1-GCaMP6s. Lineage-traced β cells are marked by mCherry in red and GCaMP6s in green. Mouse was injected IP with glucose, and video was recorded once blood glucose levels reached >200 mg/dL. Z-stacks of eight slices each 8 μm apart were recorded at ~0.2 Hz over 10 min. Scale bar is 100 μm. Time stamp shown in the upper-left corner shows time of image in min:s.
https://elifesciences.org/articles/61308#video7

lineage-tracing reporter (*Adams et al., 2018*), which uses the same Cre deriver that is used to delete *Robo* in those β cells. Thus, high expression of H2B-mCherry suggests efficient recombination of the *Robo* floxed allele. All this together support the conclusion that loss of synchronized $[Ca^{2+}]_i$ oscillations is caused by a β cell-extrinsic mechanism, such as the reduction in frequency of homotypic interactions between β cells observed in Robo βKO islets.

Robo receptors have known roles in angiogenesis and axon guidance, and thus could affect precisely how the islet is innervated and vascularized (*Blockus and Chédotal, 2016*). However, we found that the amount of innervation and vascularization between Robo βKO islets and controls is similar, suggesting that abnormal vascularization or innervation are likely not the cause of disruption in $[Ca^{2+}]_i$ oscillations found in Robo βKO islets. Additionally, our intravital imaging experiments show that the fluorescent glucose analog 2-NDBG delivered intravenously reaches all portions of Robo βKO islets within the same time frame as control islets. Thus, incomplete or delayed glucose perfusion across the islet due to improper islet vascularization is likely not the cause of asynchronous $[Ca^{2+}]_i$ oscillations in Robo βKO islets.

One possible cause for asynchronous $[Ca^{2+}]_i$ oscillations in Robo βKO islets is through disrupted coupling between β cells. Interestingly, the observed $[Ca^{2+}]_i$ oscillation phenotype in Robo βKO islets is reminiscent of the phenotype seen in heterozygous Cx36 mutants where the level of synchroneity in whole-islet $[Ca^{2+}]_i$ is highly variable from islet to islet. This suggested that Cx36-mediated gap junction coupling may be reduced in Robo βKO islets, possibly through reduction or mis-localization of Cx36. However, we found that the area of Cx36 was not different between Robo βKO and controls, and that Cx36 gap junctions still localized to the correct intracellular domain at β cell borders in Robo βKO islets. Thus, we hypothesized that coupling may be reduced instead by a reduction in the number of homotypic β cell-β cell interactions across the islet, which could reduce the overall functional connectivity. In line with this hypothesis, we found a significant reduction in β cell homotypic interactions in Robo βKO islets compared to controls, and a significant reduction in islet

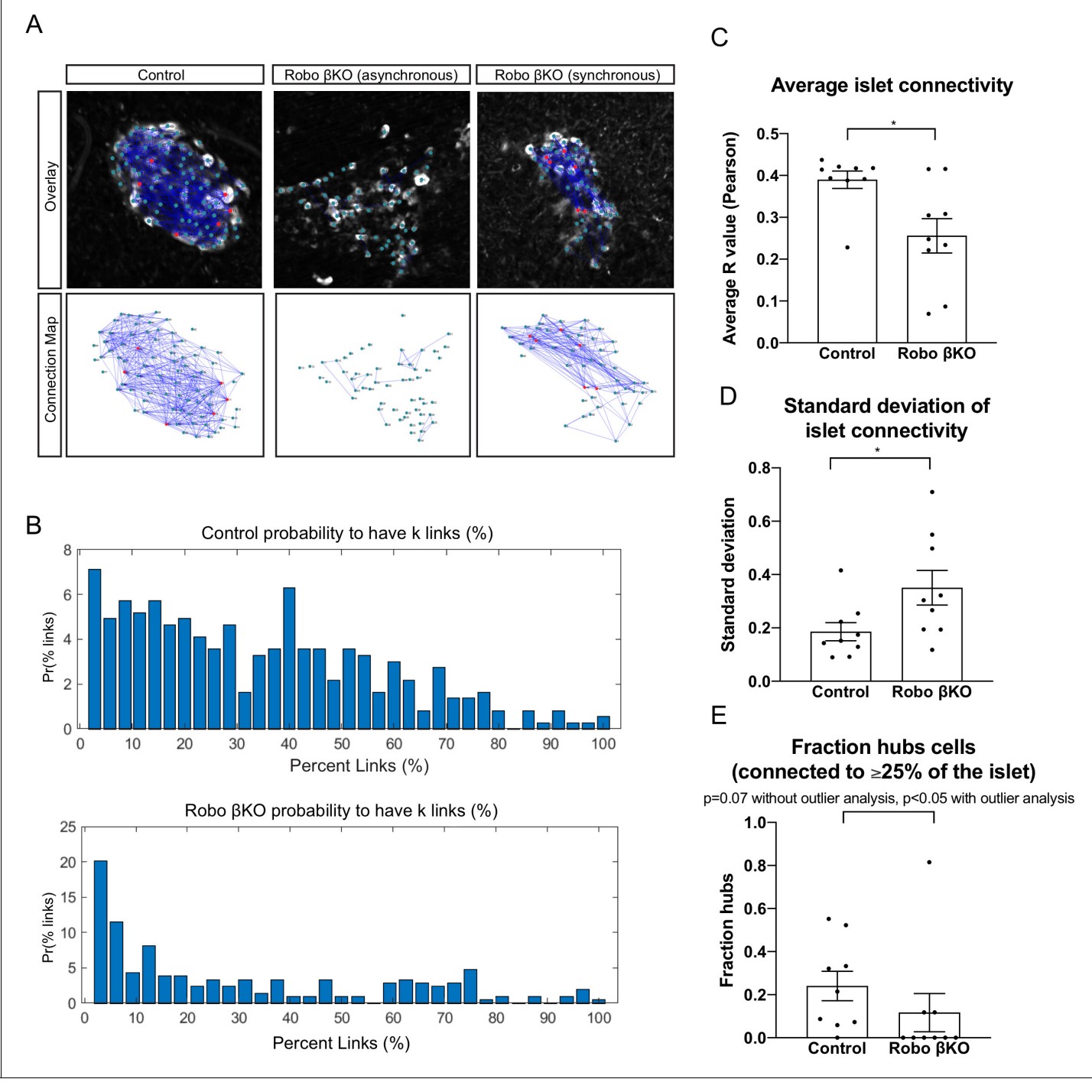

**Figure 5.** Network dynamics differ in Robo βKO islets compared to controls. (**A**) Connectivity maps of control (left panel top and bottom), asynchronous Robo βKO (middle panel top and bottom), and synchronous Robo βKO (right panel top and bottom) islets. Nodes are connected if they have a correlation coefficient of $R \geq 0.95$. Red nodes indicate cells that are connected to $\geq 25\%$ of the islet. (**B**) Probability distributions of average percent links (%) per cell within control (top) and Robo βKO (bottom) islets. (**C**) Average islet connectivity as shown by average pairwise correlation coefficient in an islet, control $R_{avg} = 0.40 \pm 0.02$ SEM, Robo βKO $R_{avg} = 0.26 \pm 0.04$, $p<0.05$ MW. (**D**) Standard deviation of $R_{avg}$ for each islet, control $0.19 \pm 0.03$, Robo βKO $0.35 \pm 0.07$, $p<0.05$ t-test. (**E**) Fraction of cells that were connected to $\geq 25\%$ of the islet (hubs) in control and Robo βKO islets, without outlier analysis control $0.24 \pm 0.07$, Robo βKO $0.12 \pm 0.09$, $p=0.07$ MW; with outlier analysis (ROUT Q = 0.1%) control $0.24 \pm 0.07$, Robo βKO $0.03 \pm 0.02$, $p<0.05$ MW. n = 9 islets each from control and Robo βKO. Error bars shown are SEM. MW: Mann–Whitney; SEM: standard error of the mean.

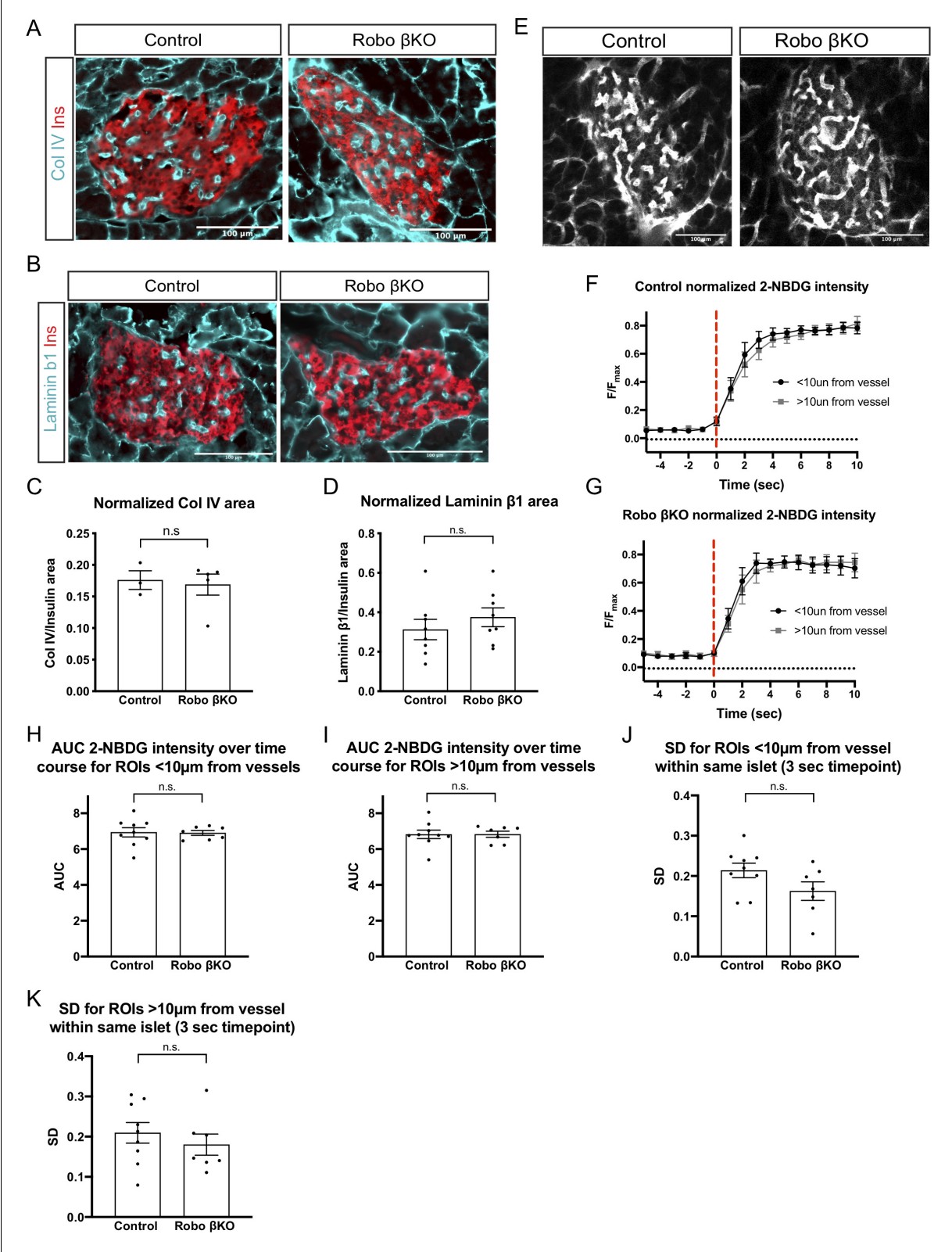

**Figure 6.** Amount and perfusion of vascularization remains unchanged in Robo βKO islets. (**A**) Representative immunofluorescent staining of collagen IV marking vasculature showing similar amounts in Robo βKO and control islets. (**B**) Representative immunofluorescent staining of laminin marking vasculature showing similar amounts in Robo βKO and control islets. (**C**) Quantification of area of collagen IV staining normalized to islet area showing no difference in amounts of basement membrane marking blood vessels in Robo βKO compared to control islets. Control n = 3 mice, Robo βKO n = 5

*Figure 6 continued on next page*

*Figure 6 continued*

mice, control 0.17 ± 0.02 SEM, Robo βKO 0.17 ± 0.02 SEM, p=0.99 MW. >12 islets from an individual mouse were measured as technical replicates, and the average values per mouse were used as biological replicates. (D) Quantification of area of laminin staining normalized to islet area showing no difference in amounts of basement membrane marking blood vessels in Robo βKO compared to control islets. Control n = 8 mice, Robo βKO n = 8 mice, control 0.31 ± 0.05 SEM, Robo βKO 0.38 ± 0.05 SEM, p=0.39 t-test. >12 islets from an individual mouse were measured as technical replicates, and the average values per mouse were used as biological replicates. (E) 2-NBDG perfused through the islet vasculature in a control and Robo βKO islet. (F) Average normalized 2-NBDG intensity (F/F$_{max}$) over time in areas < and >10 μm from nearest vessel in control islets. (G) Average normalized 2-NBDG intensity (F/F$_{max}$) over time in areas < and >10 μm from nearest vessel in Robo βKO islets. (H) Area under the curve (AUC) of 2-NBDG intensity over time in areas <10 μm from vessels in control and Robo βKO islets. Control 6.9 ± 0.26 SEM, Robo βKO 6.9 ± 0.14 SEM, p>0.90 t-test. (I) AUC of 2-NBDG intensity over time in areas >10 μm from vessels in control and Robo βKO islets. Control 6.8 ± 0.24 SEM, Robo βKO 6.8 ± 0.17 SEM, p>0.90 t-test. (J) Standard deviation of areas <10 μm from vessels within the same islet at the 3 s time point in control and Robo βKO islets. Control 0.21 ± 0.02 SEM, Robo βKO 0.16 ± 0.02 SEM, p=0.10 t-test. (K) Standard deviation of regions of interest >10 μm from vessels within the same islet at the 3 s time point in control and Robo βKO islets. Control 0.21 ± 0.03 SEM, Robo βKO 0.18 ± 0.03 SEM, p = 0.44 t-test. Error bars shown are SEM. MW: Mann–Whitney; SEM: standard error of the mean.

The online version of this article includes the following figure supplement(s) for figure 6:

**Figure supplement 1.** Robo βKO shows normal innervation within the islet.

connectivity as determined by network analysis. We also observed a significant increase in the standard deviation of average islet connectivity in our network analysis, which indicates that cells within Robo βKO islets show more variability in their level of functional connections than controls. We further hypothesized that if homotypic β cell coupling is disrupted across the islet by a decrease in frequency of homotypic β cell interactions, it could possibly affect the number of hub cells within Robo βKO islets. Since hub cells are thought to direct synchronous [Ca$^{2+}$]$_i$ oscillations among β cells within an islet, it is conceivable that disruption of this subpopulation would affect synchronous GSIS. Indeed, our network analysis revealed a trend towards a lower percent of hub cells found in Robo βKO islets compared to controls that reached statistical significance after outlier analysis. Further, we found that the average clustering coefficient (C$_{avg}$) and global efficiency (E$_{glob}$) of the networks in Robo βKO islets trended towards being lower than controls (p=0.05 for both), which is indicative of a reduction in small world properties of the network. Taken together, these results suggest that reduction in frequency of homotypic interactions reduces overall coupling and increases the variability of coupling between β cells in the islet, thus disrupting synchronous [Ca$^{2+}$]$_i$ oscillations, possibly in part due to a reduction in hub cells and loss of small world network properties. However, a direct comparison between amount of homotypic interactions and the degree of gap junction coupling or synchroneity in [Ca$^{2+}$]$_i$ oscillations within the same islet remains to be tested as this task is beyond the reach of the intravital imaging system used in our experiments. With the advent of the acute pancreatic slice method, this should be feasible as live measurements like [Ca$^{2+}$]$_i$ imaging and fluorescence recovery after photobleaching (FRAP) analysis to test gap junction coupling, and *post hoc* staining of fixed slices for architectural analysis of the same slice are possible (*Marciniak et al., 2014*). An important future experiment would thus be to test the level of gap junction coupling and synchroneity of [Ca$^{2+}$]$_i$ oscillations, along with the frequency of homotypic interactions within the same islet to directly

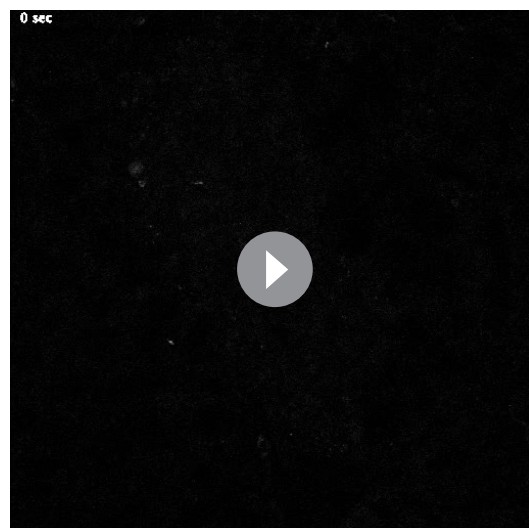

**Video 8.** Control islet perfused with the fluorescent glucose analog 2-NBDG. Intravital time-course video of an islet within the in vivo pancreas of a control mouse at 1 Hz imaging speed. Mouse was injected through tail vein IV with the fluorescent glucose analog 2-NBDG (gray) while imaging to visualize glucose analog perfusion throughout the islet. Time stamp shown in the upper-left corner shown in seconds.
https://elifesciences.org/articles/61308#video8

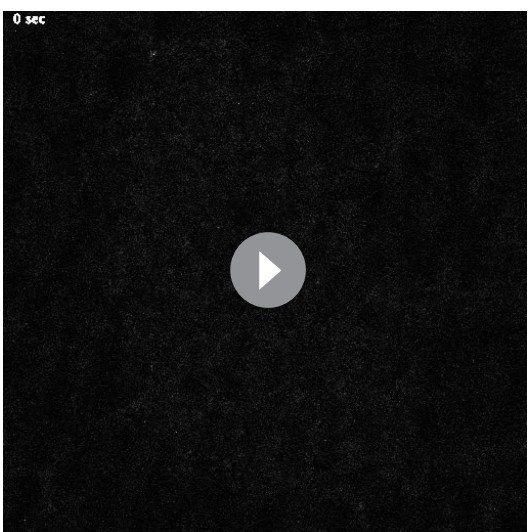

**Video 9.** Robo βKO islet perfused with the fluorescent glucose analog 2-NBDG. Intravital time-course video of an islet within the in vivo pancreas of a Robo βKO mouse at 1 Hz imaging speed. Mouse was injected through tail vein IV with the fluorescent glucose analog 2-NBDG (gray) while imaging to visualize glucose analog perfusion throughout the islet. Time stamp shown in the upper-left corner shown in seconds.
https://elifesciences.org/articles/61308#video9

correlate frequency of β cell homotypic contacts to the level of coupling and synchronous oscillatory behavior using this method.

Another possible factor that contributes to the aberrant $[Ca^{2+}]_i$ oscillations in Robo βKO islets is disruption of paracrine and autocrine signaling between endocrine cells, caused by the increased heterotypic and decreased homotypic β cell interactions in these islets. The spatial islet architecture dictates the types and amounts of paracrine and autocrine interactions among the different cell types, and these interactions are thought to regulate synchronous GSIS (*Benninger and Hodson, 2018*). For example, α cells and δ cells, which secrete glucagon and somatostatin respectively, were shown to affect cAMP oscillations in β cells, which are important for synchronizing β cell $Ca^{2+}$ oscillations during the second phase of insulin secretion (*Hodson et al., 2014*; *Tian et al., 2011*; *Grapengiesser et al., 2003*). β cells may also be able to electrically couple to neighboring δ cells, and depolarization of β cells may thus trigger secretion of somatostatin from connected δ cells, which in turn could affect islet $Ca^{2+}$ and electrical dynamics (*Briant et al., 2018*). Additionally, ATP, which is co-secreted with insulin from β cells, has an autocrine effect such that it binds

purinergic receptors on β cells to help synchronize electrical and $Ca^{2+}$ dynamics (*Hellman et al., 2004*; *Tudurí et al., 2008*). Moreover, receptor-ligand interactions occurring between neighboring β cell have also been shown to affect GSIS independent of electrical coupling. This is demonstrated by the fact that dissociated wildtype β cells have worse GSIS than intact islets lacking Cx36 gap junctions (*Benninger et al., 2011*). EphA-ephrinA binding between β cells has been identified as a mechanism that can govern this coupling-independent enhancement of GSIS (*Konstantinova et al., 2007*). These interactions rely on β cell-β cell contact and thus could be disrupted if the ratio of homotypic interactions in the islet were reduced; however, this particular effect is likely downstream of $[Ca^{2+}]_i$ oscillations and thus less likely to contribute to the oscillation phenotype observed in Robo βKO islets (*Benninger et al., 2011*). Ultimately, the spatial arrangement of endocrine cell types within the islet dictates the extent to which these paracrine, autocrine, and cell surface receptor-ligand interactions occur within the microenvironment, and thus changing this architecture would likely affect emergent electrical and $Ca^{2+}$ dynamics to some degree. Thus, while we propose that defects in synchronous $Ca^{2+}$ oscillation in response to glucose in Robo βKO islets are largely due to inefficient β cell coupling as a result of decreased homotypic β cell interactions in these islets (*Farnsworth et al., 2014*), it is also possible that changes in paracrine and autocrine signaling, as well as cell surface receptor-ligand interactions, contribute to this phenomenon.

Several patterns of $[Ca^{+2}]_i$ oscillations have been described in islets in vitro, in both isolated islets and acute pancreas tissue slices: slow oscillations with a period between ~3 and 5 min, fast oscillations with periods <1 min, and mixed oscillations consisting of fast oscillations superimposed on the plateau fraction of slow oscillations (*Ravier et al., 2002*). The Dual Oscillator Model proposes that these slow $Ca^{2+}$ oscillations are controlled by intrinsic glycolytic oscillations and set the period for pulsatile insulin secretion, whereas the fast oscillations result from feedback of ions on their respective channels and control the amount of insulin secreted by dictating the plateau fraction of the slow oscillations. Thus it is of interest if these same oscillation patterns observed in vitro in wildtype islets also occur in vivo. We observed fast, slow, and mixed oscillations in vivo in control islets, with the most common pattern being mixed and the least common being fast alone, demonstrating that these patterns are not an artifact of in vitro $Ca^{2+}$ imaging but occur in vivo in response to IP glucose

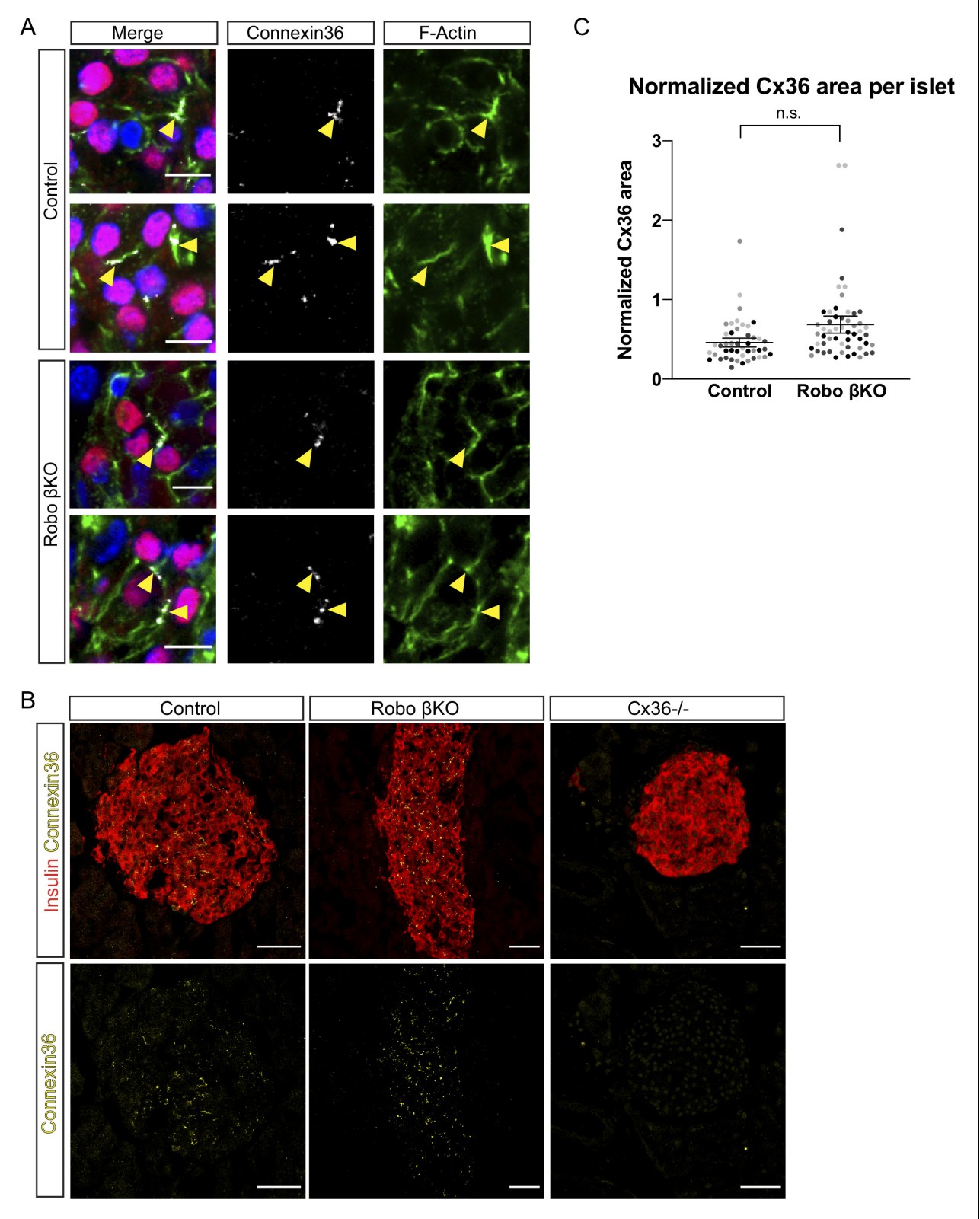

**Figure 7.** Amount of Cx36 gap junctions remains unchanged in Robo βKO. (**A**) Immunofluorescent images showing DAPI in blue, histone H2B-mCherry β cell lineage trace in red, F-actin (phalloidin) in green, and Cx36 in gray demonstrating normal localization of Cx36 to plasma membrane of β cells in both control and Robo βKO islets. Arrowheads point to colocalized F-actin and Cx36. (**B**) Immunofluorescent images of Cx36 (yellow) and insulin (red) in islets from pancreatic sections of Control, Robo βKO, and *Cx36 KO* mice. (**C**) Quantification of area of Cx36 staining normalized to islet area in Robo

*Figure 7 continued on next page*

*Figure 7 continued*

βKO islets and controls showing no significant difference, n = 4 mice for each genotype, control 0.46 ± 0.06 SEM, Robo βKO 0.69 ± 0.11 SEM, p = 0.11 t-test. Similar shaded points in graphs indicate islets from the same mouse, while mean and error bars represent statistics performed on average values from each mouse. Error bars shown are SEM. 10–14 islets were measured per mouse as technical replicates, and the average for each mouse was considered a biological replicate. SEM: standard error of the mean.

injection (*Supplementary file 2*). Slow oscillations in control islets with >80% correlated area had periods ranging from ~1 min to as long as ~10.3 min. For fast oscillations, periods ranged from ~4 s to at least ~20 s and possibly as slow as ~31 s though this high end of the range may be inflated due to possible undersampling from islets imaged at ~0.1 Hz. However, we chose to include islets imaged at ~0.1 Hz in this table because the slow oscillation speed is accurately measured and this imaging speed still confirms the presence of fast oscillations and thus provides information about oscillation pattern (i.e., mixed vs. slow). Interestingly, islets coming from the same mouse tended to have more similar oscillation periods, a property which has been observed in vitro as well (*Nunemaker et al., 2005*). Unfortunately, due to the restraints imposed by our current intravital imaging protocol, we cannot know the exact duration of and/or concentration of glucose exposure within the islet during imaging.

Though the unsynchronized phenotype in Robo βKO islets is clear from the videos we captured at approximately 1 Hz, 0.2 Hz, 0.1 Hz, and 0.03 Hz, we note that other recent experiments that analyzed network dynamics in islets were performed at an imaging speed of 1 Hz or faster. That said, we believe that it is important to capture $[Ca^{2+}]_i$ dynamics in 3D in order to ensure we are not biased by examining certain 'synchronized regions' that would not be representative of the islet dynamics as a whole. We would further argue that the slow oscillations we observe are on a ~1–10 min time scale, and therefore sampling these oscillations even at a frequency of 1 frame per 10s of seconds is sufficient for capturing synchronicity. Additionally, when we imaged at 1 Hz, we observed the same pattern of oscillations in Robo βKO islets and controls as was observed in slower imaging speeds (*Figure 4—figure supplement 2*). Furthermore, our analyses of network dynamics in simulated islets showed that functional connections and phase lag can still be accurately measured with time resolution as low as 0.1 Hz (the slowest we simulated), with only a modest loss in sensitivity (data not shown). Thus, though our imaging speeds may cause underestimation of loss in coordinated area or functional connections, it is still sensitive enough to observe the robust difference in network dynamics between Robo βKO islets and controls at these imaging speeds.

We also note that the mosaic GCaMP6s expression, which results from infection of AAV8-RIP-GCaMP6s, is variable between islets and does not allow monitoring of $Ca^{2+}$ dynamics in all β cells across the islet. However, the distribution of GCaMP6s in these islets is similar if not broader than what is achievable with incubation of commonly used $Ca^{2+}$ dyes that have proven successful in measuring $Ca^{2+}$ dynamics in the islet. Specifically, the AAV8 we used penetrates into the core of the islets instead of just the superficial layers as is often seen with $Ca^{2+}$ dye incubation, allowing for equal if not greater representation of the islet in imaging. Thus, though GCaMP6s expression is mosaic and variable between islets, the expression is comparable to other common methods that have been useful in assessing $Ca^{2+}$ dynamics. However, future experiments would benefit from using the genetically encoded GCaMP6s floxed allele combined with a β cell-specific Cre driver to increase whole-islet resolution of $Ca^{2+}$ wave dynamics (*Madisen et al., 2015*).

Taken together, our data thus supports the hypothesis that disrupting proper endocrine cell-type sorting in the islet in a way that distorts the relative amount of homotypic β cell-β cell contacts, without affecting β cell-intrinsic $[Ca^{2+}]_i$ dynamics, is sufficient to disrupt synchronized $[Ca^{2+}]_i$ oscillations among β cells. These results lend evidence to the idea that correct spatial islet architecture is important for islet function, and may have an important impact on understanding islet dysfunction in diabetes, and on approaches to generate functionally better islets from stem cells in vitro.

# Materials and methods

**Key resources table**

| Reagent type (species) or resource | Designation | Source or reference | Identifiers | Additional information |
|---|---|---|---|---|
| Genetic reagent (*Mus musculus*) | *Robo1*<sup></sup>*tm1Matl* ;*Robo2*<sup></sup>*tm1Rilm* | PMID:26743624 | RRID:MGI:3043177 RRID:MGI:5705332 | Linked alleles provided by Xin Sun |
| Genetic reagent (*M. musculus*) | *Ins1*<sup></sup>*tm1.1(cre)Thor* | Jackson Laboratory | RRID:IMSR_JAX:026801 | |
| Genetic reagent (*M. musculus*) | B6.FVB(Cg)-Tg(*Ucn3-cre*)KF43Gsat/Mmucd | MMRRC | RRID:MMRRC_037417-UCD | |
| Genetic reagent (*M. musculus*) | R26H2BCherry | PMID:25233132 | R26H2BCherry | Generated as described in source |
| Genetic reagent (*AAV8*) | VB171020-1118pud | PMID:31186447 Generated by VectorBuilder | AAV8-INS-GcAMP6s | Sequence available in Supplementary materials of source |
| Antibody | Anti-insulin (guinea pig polyclonal) | Agilent | Cat#:IR002 RRID:AB_2800361 | (1:6) |
| Antibody | Anti-glucagon (mouse monoclonal) | Sigma | Cat#:G2654 RRID:AB_259852 | (1:500) |
| Antibody | Anti-glucagon (rabbit polyclonal) | Cell Signaling | Cat#:2760S RRID:AB_659831 | (1:200) |
| Antibody | Anti-somatostatin (rabbit polyclonal) | Phoenix | Cat#:G-060-03 RRID:AB_2890901 | (1:1000) |
| Antibody | Anti-Cx36 (rabbit polyclonal) | Invitrogen | Cat#:36-4600 RRID:AB_2314259 | (1:80) |
| Antibody | Anti-Col IV (rabbit polyclonal) | Abcam | Cat#:Ab6586 RRID:AB_305584 | (1:300) |
| Antibody | Anti-laminin β1 (rat monoclonal) | Invitrogen | Cat#:MA5-14657 RRID:AB_10981503 | (1:500) |
| Antibody | Anti-Tubb3 (rabbit polyclonal) | BioLegend | Cat#:poly18020 RRID:AB_2564645 | (1:4000) |
| Antibody | Anti-guinea pig 594 (donkey polyclonal) | Jackson | Cat#:706-585-148 RRID:AB_2340474 | (1:500) |
| Antibody | Anti-guinea pig 647 (donkey polyclonal) | Jackson | Cat#:706-605-148 RRID:AB_2340476 | (1:500) |
| Antibody | Anti-rabbit 594 (donkey polyclonal) | Invitrogen | Cat#:A21207 RRID:AB_141637 | (1:500) |
| Antibody | Anti-goat 647 (donkey polyclonal) | Invitrogen | Cat#:A21447 RRID:AB_141844 | (1:500) |
| Antibody | Anti-rat 488 (donkey polyclonal) | Invitrogen | Cat#:A21208 RRID:AB_141709 | 1:500 |
| Antibody | Anti-rabbit 488 (donkey polyclonal) | Invitrogen | Cat#:A21206 RRID:AB_2535792 | (1:500) |
| Antibody | Anti-mouse 488 (donkey polyclonal) | Jackson | Cat#:715-546-151 RRID:AB_2340850 | (1:500) |
| Antibody | Anti-biotin 488 (mouse polyclonal) | Jackson | Cat#:200-542-211 RRID:AB_2339040 | (1:500) |
| Chemical compound, drug | Phalloidin-647 | Invitrogen | A22247 | (1:400) |
| Chemical compound, drug | 2-NBDG | Invitrogen | N13195 | (5 mg/mL) |
| Chemical compound, drug | Fura2-AM | Thermo Fischer | F1201 | (5 µm) |
| Peptide, recombinant protein | Accutase | Thermo Fischer | A1110501 | |
| Peptide, recombinant protein | Geltrex | Thermo Fischer | A1413302 | |

*Continued on next page*

*Continued*

| Reagent type (species) or resource | Designation | Source or reference | Identifiers | Additional information |
| --- | --- | --- | --- | --- |
| Software, algorithm | Prism | GraphPad | RRID:SCR_005375 | |
| Software, algorithm | MATLAB | MathWorks | RRID:SCR_001622 | |

## Animals

The experimental protocol for animal usage was reviewed and approved by the University of Wisconsin-Madison Institutional Animal Care and Use Committee (IACUC) under Protocol #M005221 and Protocol #M005333, and all animal experiments were conducted in accordance with the University of Wisconsin-Madison IACUC guidelines under the approved protocol. *Robo1$^\Delta$,Robo2$^{flx}$* (*Branchfield et al., 2016*), *Ins1-Cre* (*Thorens et al., 2015*), *Urocortin3-Cre* (*van der Meulen et al., 2017*), and *Rosa26-Lox-Stop-Lox-H2BmCherry* (*Blum et al., 2014*) mice were previously described. All mouse strains were maintained on a mixed genetic background. Control colony mates in all analyses were *Robo$^{+/+}$* with either *Ins1-Cre or Ucn3-Cre*.

## Immunofluorescence

Pancreata were fixed with 4% PFA at 4°C for 3 hr, embedded in 30% sucrose, and frozen in OCT (Tissue-Tek). Pancreatic sections (10 μm) were stained using a standard protocol. The following primary antibodies and dilutions were used: guinea pig anti-insulin (1:6, Dako, IR00261-2), mouse anti-glucagon (1:500, Sigma G2654), rabbit anti-glucagon (1:200, Cell Signaling 2760S), rabbit anti-somatostatin (1:1000, Phoenix G-060-03), rabbit anti-Cx36 (1:80, Invitrogen 36-4600), rabbit anti-Col IV (1:300, Abcam Ab656), rat anti-laminin β1 (1:500, Invitrogen MA5-14657), and rabbit anti-Tubb3 (1:4000, BioLegend poly18020). The following secondary antibodies were used at 1:500: donkey anti-guinea pig 594 (Jackson), donkey anti-guinea Pig 647 (Jackson), donkey anti-rabbit 488 (Invitrogen), donkey anti-rabbit 594 (Invitrogen), donkey anti-goat 647 (Invitrogen), donkey anti-rat 488 (Invitrogen), donkey anti-mouse 488 (Jackson), and donkey anti-biotin 488 (Jackson). Slides were imaged using a Leica SP8 Scanning Confocal microscope or a Zeiss Axio Observer.Z1 microscope.

## RNA sequencing

RNA was isolated from FACS sorted lineage-traced β cells (*Adams et al., 2018*) from control and Robo βKO mice using phenol chloroform extraction (TRIzol). DNA libraries were generated using Takara's SMART-Seq v4 Low Input RNA Kit for Sequencing (Takara, Mountain View, CA) for cDNA synthesis and the Illumina NexteraXT DNA Library Preparation (Illumina, San Diego, CA) kit for cDNA dual indexing. Full-length cDNA fragments were generated from 1 to 10 ng total RNA by SMART (Switching Mechanism at 5′ End of RNA Template) technology. cDNA fragments were fragmented and dual indexed in a single step using the Nextera kit's simultaneous transposon and tagmentation step. Quality and quantity of completed libraries were assessed using Agilent DNA series chip assay (Agilent Technologies, Santa Clara, CA) and Invitrogen Qubit ds DNA HS Kit (Invitrogen, Carlsbad, CA), respectively. Each library was standardized to 2 nM. Cluster generation was performed on Illumina cBot, with libraries multiplexed for 1 × 100 bp sequencing using TruSeq 100 bp SBS kit (v4) on an Illumina HiSeq2500. Images were analyzed using standard Illumina Pipeline, version 1.8.2.

## Intravital imaging for dynamic [Ca$^{2+}$]$_i$

Mouse pancreata were exposed in anesthetized mice by making a small incision on the right side of the mouse and externalizing the tip of the pancreas. A glass dish was placed over the exposed pancreas and the mouse was placed on microscope stage over an inverted objective with isoflurane anesthesia for the remainder of imaging. Islets were identified on the surface of the pancreas by detecting Histone H2B-mCherry fluorescent nuclei labeled by a β cell-specific lineage-tracing reporter (*Adams et al., 2018*). Once islets were identified, mice were given injections of 1 g/kg body weight glucose (30% in saline) intraperitoneally. Blood glucose levels were monitored through tail vein bleeds. Once the blood glucose reached at least 200 mg/dL, GCaMP6s activity was identified using the microscope eye piece. Time courses of GCaMP6s were imaged using a custom-built

inverted multiphoton microscope (Bruker Fluorescence Microscopy, Middleton, WI), as described previously (*Lugo-Cintrón et al., 2020*). The system consisted of a titanium:sapphire laser (Spectra-Physics, Insight DS-Dual), an inverted microscope (Nikon, Eclipse Ti, Melville, New York, NY), and a Nikon Apo 20x objective. For recordings, z-stacks were set to 1, 3, 8, or 12 slices each 8 µm apart and images were captured at 1 Hz, 0.2 Hz, 0.1 Hz, or 0.03 Hz, respectively, over at least 10 min at a resolution of 512 × 512 pixels. GCaMP6s and mCherry signal were excited with a laser tuned to 890 or 910 nm. After time courses were recorded, high-resolution image z-stacks were taken with 60 z-planes taken 1 µm apart or 8 z-planes taken 8 µm apart at 1024 × 1024 pixel resolution. For some images, 70 kDA rhodamine-dextran was injected retroorbitally to mark the vasculature of the islets in vivo.

## Gap junction, vasculature, and innervation quantification

Cx36 levels were quantified from images of islets co-stained with rabbit anti-Cx36 (Invitrogen) and guinea pig anti-insulin (Dako) antibody. Vasculature levels were quantified from images co-stained with rat anti-laminin β1 (Invitrogen) or rabbit anti-col IV and guinea pig anti-insulin (Dako). Innervation was quantified from images of tissue sections co-stained with rabbit anti-Tubb3 (BioLegend) and guinea pig anti-insulin (Dako). For vasculature and Cx36, eight Z-planes were taken 1 µm apart, and for innervation, five z-planes were taken 1 µm apart, on a Leica SP8 Scanning Confocal microscope using a 40× oil immersion objective (Cx36) or 20× objective (vasculature and innervation). Filtering and thresholding were applied to both channels for each islet, and the area of each staining was measured using Fiji's analyze particles functions. The area of gap junctions, blood vessels, or innervation (marked by their respective antibody) was divided by the area of DAPI (for Cx36) or insulin (vasculature and innervation) for each islet.

## Intravital imaging and analysis of 2-NBDG perfusion

Mouse pancreata were exposed in anesthetized mice by making a small incision on the right side of the mouse, and externalizing the tip of the pancreas, and a tail vein catheter was inserted. A glass dish was placed over the exposed pancreas and the mouse was placed on a microscope stage over an inverted objective with isoflurane anesthesia for the remainder of imaging. Intravital imaging was performed on a custom-built multi-photon microscope as described above. Islets were identified on the surface of the pancreas by detecting Histone H2B-mCherry fluorescent nuclei labeled by β cell-specific lineage-tracing reporter (*Adams et al., 2018*). Recordings were taken of a single plane within the islet at 1 Hz imaging speed and 512 × 512 pixel resolution using a 20× objective and the laser tuned to 820 nm. During recording, 50 µL of 5 mg/mL 2-NBDG diluted in PBS (Invitrogen N13195) were injected through tail vein catheter, with recordings continuing for 5 min after injection.

Perfusion of 2-NBDG was measured in the pancreas using the ImageJ Time Series Analyzer Plugin with the following workflow. A 10 × 10 µm nondestructive grid was overlaid on each movie and the time point with the most inter-vascular 2-NBDG intensity was used to determine blood vessel location in the islet. 5 µm diameter circular regions of interest (ROIs) were added randomly across the islet in areas within 10 µm of vessels and areas >10 µm away from the closest vessel as determined by overlaid grid. Average intensities of ROIs were then measured over the time span 5 s before and 10 s after the frame with the first >5 unit increase of intensity in the 2-NBDG channel averaged over the entire image, marking the entry of 2-NBDG into the islet. The average intensity for each ROI was then normalized to basal-peak intensity ($F/F_{max}$) by subtracting the minimum intensity over the −5 to 10 s time frame from each time point and then multiplying those values by 1/(max intensity − min intensity). The normalized average intensities for all ROIs within the same islet within 10 µm and >10 µm from vessel ROIs were then averaged respectively to give a value for average 2-NBDG perfusion close to and far away from vessels for each islet at each time point. The AUC was measured using Prism (GraphPad) for the normalized average 2-NBDG intensity over −5 to 10 s for regions within 10 µm and >10 µm for each islet.

## Nearest-neighbor analysis

β cells were identified using the lineage tracer *Rosa26-Lox-Stop-Lox-H2BmCherry* crossed to *Ucn3-Cre,* and tissue sections were stained with antibodies against glucagon and somatostatin to identify

α and δ cells, respectively. The 3D Tissue Spatial Analysis Toolbox for Fiji (*Tran Thi Nhu et al., 2017*) was used to identify specific cell types using the above markers and calculate the number of cell-type-specific nearest neighbors from all identified endocrine cells.

## Correlated area, wave speed, and phase lag analysis

All images were analyzed using previously published methods (*Westacott et al., 2017b*) with custom MATLAB (MathWorks) scripts. Islets were defined as all mCherry+ cells that were direct neighbors with multiple other mCherry+ cells within a cluster. Islets were initially selected as showing activity through blinded selection by two separate researchers who are knowledgeable about islet $[Ca^{2+}]_i$ dynamics and who did not perform the imaging. If either researcher determined any islet to be inactive, the islet was not used for analysis. From this assessment, 16 out of 22 control islets and 18 out of 23 Robo βKO islets were determined to be active. Subsequent analysis was also performed blinded to genotype. For activity analysis, images were smoothed using a 5 × 5 pixel averaging filter. Areas without significant fluorescence were removed. Saturated areas were also removed by limiting the area to intensity below the maximum value. Photobleaching was low, and as such could be approximated as a linear decline. Thus any linear trend was removed to correct for minimal photobleaching. Any islets with significant motion artifacts were removed or time courses were shortened to the time over which no significant movement occurred (displacement of <0.5 cell width). For the time course of each pixel in the image with significant fluorescence, a peak detection algorithm was used to determine if the areas had peak amplitudes significantly above background (*Westacott et al., 2017b*). A region was considered 'active' if the corresponding time course for each pixel had a peak amplitude >1.5× background. The fraction of active area was calculated as the number of pixels detected as 'active' across all z-planes, normalized to the total number of pixels that showed significant fluorescence across all z-planes that were not saturated. Coordination was determined based on coincident timing of identified peaks, where areas were segmented by identified peaks occurring at similar time points (*Westacott et al., 2017b*). The cross-correlation of the time courses for two 5 × 5 pixel subregion was taken. If the correlation coefficient was >0.7, then the two subregions were considered highly coordinated and merged into a larger region. The coordinated area was calculated as the number of pixels in the largest area of coordination across all z-planes normalized to the total number of pixels of the islet that were determined to be 'active' for all planes. This analysis is based on previous analysis (*Westacott et al., 2017b*), but adjusted for three-dimensional data.

Phase lag and wave propagation speed was determined, as in *Benninger et al., 2008*. For every 5 × 5 region, the phase lag was calculated from a Fourier transform of each time course. Only regions with a correlation coefficient of >0.7 when compared to the islet average were used. First, the peak frequency was identified from the power spectrum, as generated from a Fourier transform of the islet average time course. The phase lag was only calculated for the peak frequency. The phase lag was calculated from the difference in phase between the maximum phase region and the minimum phase region and converted into a time lag according to ($dt = (1/f) * \tan(-1)(\phi 1 - \phi 2)$), where φ1−φ2 is the phase difference, and *f* is the sampling frequency of the time courses analyzed. Speed was calculated by dividing the distance between the maximum phase lag and minimum phase lag regions by the time lag. For phase and speed analysis, only islets with >80% coordinated area was used. If the islets did not show a clear phase transition across the islet, they were excluded from the analysis.

All statistical analysis was performed in Prism (GraphPad) or MATLAB. If the data were determined to follow a normal distribution, an F-test was used to determine if variances were equal, then a Student's t-test or Welch t-test (for unequal variance) were utilized. If a normal distribution could not be verified for the data, a Mann–Whitney test was utilized for determining significance. $p < 0.05$ was considered significant.

## Network and hub analysis

The network analysis is based upon that previously applied to study islet $Ca^{2+}$ dynamics (*Johnston et al., 2016*; *Stožer et al., 2013*). First, a single plane in the z-stack was chosen that showed the greatest coverage of GCaMP6s signal. Visible cells were manually identified and the $Ca^{2+}$ time courses for each cell were derived. To remove bias, an equal number of islets from each

temporal resolution were used for both the control and knockout. There was no difference in results between islets imaged at a speed of 0.1, 0.2, and 1 Hz. Adjacency matrices for an islet were created using the corr() function in MATLAB to compare the entire time course between each cell and calculate the Pearson's correlation coefficient between cells in a pairwise manner. Diagonals were set to 0 as cells are not considered synchronized with themselves. The control group was used to set the R_th threshold (as set forth by *Stožer et al., 2013*). All control time courses showed a power law with R_th = 0.98. However, the threshold for analysis was set to 0.95 because most Robo βKO islets did not show any connectivity with R_th at 0.98, which thus prevented comparisons. This threshold was statistically significant for p<0.001 for all islets.

To identify β cell hubs, a probability distribution was created by counting the number of cells that had X number of links, for all values of X. Cells with zero links were not plotted in the histogram but were considered in the subsequent analysis. Hubs were identified based on the percent of cells that are synchronized (linked) being more than 25% of the islet. This allows for size invariance and consistency for comparison between control and KO groups. In contrast in the methods set forth in *Johnston et al., 2016*, hubs are considered cells with more than 60% of the islet's connections. The number of hubs was then normalized by islet size. This is dependent on the individual islet probability distribution function, and inter-islet comparison is contingent on the fact that the islets follow the same network distribution. This is not necessarily the case when comparing control and Robo βKO islets.

The clustering coefficient was calculated as the number of connections for cell(i) divided by the number of possible connections. Because no constraints were placed on possible connections, this was the total number of cells in the islet – 1. The average clustering coefficient presents the mean of C for all cells.

Global efficiency was calculated as follows. First, shortest path length was calculated with MAT-LAB function graphallshortestpaths(). This function uses the Johnston et al. algorithm (*Johnston et al., 2016*) to find the shortest path between every pair of cell in the islet. For example, the path length between cell(i) and cell(j) is 1 if they are directly synchronized, or 2 if cell(i) is not synchronized with cell(j), but each is synchronized with cell(k). Cells without any path connecting them were not considered. Finally, the characteristic path length (L) was calculated by summing up all non-zero path lengths normalized to total possible connections (size * (size-1)). The global efficiency is related to the inverse of global path length (*Latora and Marchiori, 2001*):

$$E_{\{global\}} = \frac{1}{Islet_{size} * (Islet_{size} - 1)} \sum \frac{1}{L_{j,k}}$$

## In vitro single-cell Ca²⁺ imaging

Islets were isolated according to standard protocol from adult Robo βKO and control mice. For islet dispersion, 12 mm round No. 1.5 coverslips contained in a 24-well plate were pre-coated overnight with 50 µL 1:15000 PEI (Sigma P3143) overnight. Groups of 50–100 mouse islets were dispersed into single cells in 3 mL Accutase (Thermo Fisher A1110501) at 37°C for 10 min. During the incubation, PEI was replaced with 100 µL Geltrex (Thermo Fisher A1413302) and centrifuged at 500 g for 5 min at 4°C, followed by removal of excess Geltrex. The cells were washed once with islet culture medium (RPMI1640 supplemented with 10% FBS [v/v], 100 units/mL penicillin, and 100 µg/mL streptomycin [Invitrogen]) and resuspended in 1 mL medium before plating 500 µL per coverslip. The plate was centrifuged for 5 min at 500 g and cultured overnight before imaging. For measurements of cytosolic Ca²⁺, dispersed islet cells were pre-incubated in 5 µM Fura2-AM (Thermo Fisher F1201) in islet media containing 11.1 mM glucose for 45 min at 37°C, followed by 15 min incubation in islet media containing 2.7 mM glucose. Coverslips were transferred to a RC-48LP imaging chamber (Warner Instruments) mounted on a Nikon Ti-Eclipse inverted microscope equipped with a 20×/0.75N.A. SuperFluor objective and PerfectFocus (Nikon Instruments). The chamber was perfused with a standard external solution containing 135 mM NaCl, 4.8 mM KCl, 2.5 mM CaCl₂, 1.2 mM MgCl₂, 20 mM HEPES, and glucose as indicated (pH 7.35). The flow rate was set to 0.4 mL/min (Fluigent MCFS-EZ) and temperature was maintained at 33°C using solution and chamber heaters (Warner Instruments). Excitation was provided by a SOLA SE II 365 (Lumencor) set to 10% output and an inline neutral density filter (Nikon ND4). Fluorescence emission was collected with a Hamamatsu ORCA-Flash4.0 V2 Digital CMOS camera at 0.1 Hz. Excitation (x) and emission (m) filters were used in combination with

a ET FURA2/GFP C164605 dichroic (Chroma): Fura2, ET365/20x, ET535/30m; mCherry ET572/35x, and ET632/60m. β cells were identified by the expression of mCherry. Baseline-normalized $[Ca^{2+}]_i$ was quantified using Nikon Elements and GraphPad Prism software.

## Statistical analysis

All analyses were performed in Prism (GraphPad) unless otherwise stated. For data with normal distribution (determined visually and by Shapiro–Wilk test) and equal variance (determined by F-test), a t-test was used. For normally distributed data with unequal variance between groups, a t-test with Welch's correction was used. For non-normally distributed data, a Mann–Whitney test was used to determine significance. $p > 0.05$ was considered significant.

## Acknowledgements

We thank Kurt Weiss, Jan Huisken, and David Inman for help with imaging. We thank members of the Blum lab, especially Jennifer Gilbert and Bayley Waters, for valuable discussion. We also thank Joana Almaca and Luciana Goncalves for valuable discussions. We are also grateful to Nadav Sharon and Danny Ben-Zvi for critically reading the manuscript. This work was funded in part by the following grants. R01DK121706, the DRC at Washington University Pilot Grant P30DK020579, and Pilot Award UL1TR000427 from the UW-Madison Institute for Clinical and Translational Research (ICTR) to BB; R01DK060581 to RM; R01DK102950 and R01DK106412 to RKPB; R01CA216248 to SP; R01DK113103, R01AG062328, ADA 1-16-IBS-212 to MM, and an award from the Wisconsin Partnership Program to BB and MM. MTA was funded by 5T32GM007133-44 and a graduate training award from the UW-Madison Stem Cell and Regenerative Medicine Center. We also thank the University of Wisconsin Carbone Cancer Center Support Grant P30CA014520 for use of the UW Flow Core, the University of Wisconsin, Madison Biotechnology Center for sequencing and analysis, and the University of Wisconsin, Madison Optical Imaging Core (UWIOC) support grant 1S10OD025040-01, for use of their multiphoton microscope for preliminary imaging.

## Additional information

### Funding

| Funder | Grant reference number | Author |
| --- | --- | --- |
| National Institute of Diabetes and Digestive and Kidney Diseases | R01DK121706 | Barak Blum |
| National Institute of Diabetes and Digestive and Kidney Diseases | P30DK020579 | Barak Blum |
| Institute for Clinical and Translational Research, University of Wisconsin, Madison | UL1TR000427 | Matthew J Merrins Barak Blum |
| National Institute of Diabetes and Digestive and Kidney Diseases | R01DK060581 | Raghavendra G Mirmira |
| National Institute of Diabetes and Digestive and Kidney Diseases | R01DK102950 | Richard KP Benninger |
| National Institute of Diabetes and Digestive and Kidney Diseases | R01DK106412 | Richard KP Benninger |
| National Cancer Institute | R01CA216248 | Suzanne M Ponik |
| National Institute of Diabetes and Digestive and Kidney Diseases | R01DK113103 | Matthew J Merrins |
| National Institute on Aging | R01AG062328 | Matthew J Merrins |

| American Diabetes Association | ADA 1-16-IBS-212 | Matthew J Merrins |
| National Institute of General Medical Sciences | 5T32GM007133-44 | Melissa T Adams |

The funders had no role in study design, data collection and interpretation, or the decision to submit the work for publication.

### Author contributions

Melissa T Adams, Conceptualization, Formal analysis, Investigation, Methodology, Writing - original draft, Writing - review and editing; JaeAnn M Dwulet, Jennifer K Briggs, Formal analysis, Writing - review and editing; Christopher A Reissaus, Conceptualization, Formal analysis, Investigation, Methodology, Writing - review and editing; Erli Jin, Melissa R Lyman, Sophia M Sdao, Formal analysis, Investigation, Writing - review and editing; Joseph M Szulczewski, Investigation, Methodology, Writing - review and editing; Vira Kravets, Methodology, Writing - review and editing; Sutichot D Nimkulrat, Investigation, Writing - review and editing; Suzanne M Ponik, Raghavendra G Mirmira, Resources, Funding acquisition, Writing - review and editing; Matthew J Merrins, Richard KP Benninger, Resources, Supervision, Funding acquisition, Writing - review and editing; Amelia K Linnemann, Conceptualization, Resources, Supervision, Funding acquisition, Methodology, Writing - review and editing; Barak Blum, Conceptualization, Resources, Formal analysis, Supervision, Funding acquisition, Methodology, Writing - original draft, Writing - review and editing

### Author ORCIDs

Melissa T Adams (ID) https://orcid.org/0000-0001-5600-5562
JaeAnn M Dwulet (ID) http://orcid.org/0000-0003-2519-5193
Melissa R Lyman (ID) http://orcid.org/0000-0002-0091-0413
Vira Kravets (ID) http://orcid.org/0000-0002-5147-309X
Richard KP Benninger (ID) http://orcid.org/0000-0002-5063-6096
Barak Blum (ID) https://orcid.org/0000-0002-5308-4194

### Ethics

Animal experimentation: The experimental protocol for animal usage was reviewed and approved by the University of Wisconsin-Madison Institutional Animal Care and Use Committee (IACUC) under Protocol #M005221 and Protocol #M005333, and all animal experiments were conducted in accordance with the University of Wisconsin-Madison IACUC guidelines under the approved protocol.

### Decision letter and Author response

Decision letter https://doi.org/10.7554/eLife.61308.sa1
Author response https://doi.org/10.7554/eLife.61308.sa2

# Additional files

### Supplementary files

• Supplementary file 1. Differentially expressed transcripts linked to [beta] cell maturity and differeantion.
• Supplementary file 2. Robo βKO and control oscillation patterns.
• Transparent reporting form

### Data availability

All data generated or analyzed during this study are included in the manuscript and supporting files.

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
