## [Decision Letter]

**Acceptance summary:**

The reviewers and Editorial team all appreciated your efforts to deal with the concerns raised, and recognised the importance of this work and the insights it provides both on the function of Robo in the β cell and to the development of a sophisticated in vivo imaging technology. They also recognised your further refinements and elaboration of calcium imaging within the islet in the living animal but understood that further improvements to this approach, notably to increase the speed of data capture, would be best suited to a future study.

**Decision letter after peer review:**

Thank you for submitting your article "Loss of synchronized β cell response to glucose in Robo deficient islets of Langerhans in vivo" for consideration by *eLife*. Your article has been reviewed by 3 peer reviewers, and the evaluation has been overseen by a Reviewing Editor and David James as the Senior Editor. The following individuals involved in review of your submission have agreed to reveal their identity: Andras Stozer (Reviewer #1); Marjan Slak Rupnik (Reviewer #3).

The reviewers have discussed the reviews with one another and the Reviewing Editor has drafted this decision to help you prepare a revised submission.

The authors have used an elegant approach to image pancreatic islet calcium dynamics in vivo (whose feasibility was demonstrated recently by the Linnemann laboratory; https://pmlegacy.ncbi.nlm.nih.gov/pubmed/31186447 ) to assess the impact of Robo1 and Robo2 deletion, posited to have an effect on the islet only to change the distribution of endocrine cells within the micro-organ and therefore, conceivably, to affect calcium waves and connectivity via altered gap junction connectivity. The authors argue that the impact of Robo1/2 deletion on islet stability is so drastic that islets fall apart upon isolation, precluding in vitro studies.

Although of interest, the manuscript is quite preliminary in many respects, and new work is required to make a compelling case for publication. As detailed below, essential revisions will therefore involve: assessment of the impact of Robo1/2 deletion on Connexin distribution; the impact of deletion on gap junction coupling using dyes; and assessment of the possibility that heterogeneity in calcium wave conduction between islets reflects differences in the extent of disruption of islet architecture (degree of inter-mingling of endocrine cell types). More careful and extensive description of the literature, particularly around intercellular coupling, network behaviour etc is also.

1. The authors infer that preserved calcium waves in a proportion of Robo KO islets likely reflects preserved architecture. This needs to be checked post facto by appropriate staining of the different cell types

2. On the same theme, some ROBO islets show synchronous intra-islet regions. Moreover, some ROBO islets such as the one shown in Movie 6 are indistinguishable from WT in terms of Ca^2+^ oscillations. Are the regions/islets, which show synchronous oscillations indeed the ones with preserved architecture i.e. normal homotypic interactions of the β-cells? This can be examined by Ca^2+^ imaging followed by IHC analysis of the islets (as in Figure 1) – provided that the same islet can be monitored using imaging and then processed for IHC.

3. Acquisition speeds are very low (max 0.2 Hz) so that – as the authors accept – only very low speed Ca^2+^ waves can be detected. I am not full persuaded that imaging throughout the islet in 3D is justified. This limits the conclusions that can be drawn from the study, and comparisons to many recent reports (see 3/) where acquisition times >1 Hz are used. Surprisingly, recent and earlier reports from the laboratories of Stozer, Rupnik (see below), Hodson, JCI, 2013; Rutter, Cell Metab, 2016; Ninov, Nat Metab, 2019 on network dynamics and hub/follower behaviour are missing. The authors are rightly cautious about discussing the possibility that Robo deletion impacts these behaviours given the limitations of the data acquisition (low speed), but this work still needs to be properly cited and put into context.

Moreover, if a higher sampling reduces the measured coordination even further, than this coordination is likely not physiologically relevant. The images as collected a 512x512, but smoothed 5x5 for analysis. Would it not be better to collect 100x100 images directly and image 25x faster to check the physiological significance of coordination of oscillations? Discussion and comparisons required.

4. Limited expression of GCamP6 across the islet e.g. Supp. Figure 2 poses a problem in terms of accurately reporting Ca^2+^ wave transduction. Fuller description of this limitation and its impact are needed

5.The assessment of Cx36 expression is Figure 6 is not wholly compelling: is the protein actually localised at the plasma membrane or some other (intracellular?) site. Higher resolution light or electron microscopy are needed. The authors provide a hypothesis that preferential homotypic contact allows for a necessary amount of gap junctions to form. Observed reduction of homotypic contacts between the β cells is not associated with the reduction of Cx36 expression, and it is therefore not clear what is meant by the missing necessary amount of gap junctions? Furthermore, colocalization of Cx36 should not be considered equal to functional gap junctions, and smaller fraction of homotypic nearest neighbors in Robo betaKO β cells does not necessarily equal limited capacity to electrically couple. The question is how is the junctional conductance distributed within an islet? The published data describes a complex temporal pattern of fluctuations in junctional conductance between the β cell pairs, but overall islet junctional conductance is conserved (Jusup et al., 2020, Biophys J).

6. The claim that Gap junctions are properly formed in ROBO KO islets needs substantiation using a functional assay. For example, the authors should apply fluorescence recovery after photobleaching to assess GJ function, as described before by Farnsworth et al. (PMID: 25172942), or other means to monitor GJ permeability.

7. In the Introduction, the view that mouse/rodent islets are of the classical mantle-core type may be a bit too simplified. I suggest that the authors consider/point out the possibility that at least some of the islets, at least during some normal/pathological developmental stages may display other architectures, of which some may be close to the more complex human architecture. I strongly believe that the study could benefit from this addition, since the authors strongly argue for the importance of structure in determining function. In other words, architectural patterns deviating from the textbook core-mantle paradigm may indicate important adaptations to normal (pregnancy, age) or pathological (hyperglycemia) stimuli. For sure, this is something that adds to the importance of their study. Some of the additional useful references are for example:

Dolensek et al., Islets 2015.

Steiner et al., Islets 2010.

Orci, Lancet 1975.

Pfeifer, Struct Biol 2015.

Erlandsen, Histochem Cytochem 1976.

Kharouta, Diab Res Clin Pract 2009.

8. In the introduction, gap-junctional coupling is pointed out as the only important synchronizing mechanism affected by homotypical interactions/cell-cell contacts. A number of other structural and functional coupling mechanism have been proposed to operate in islets and practically all of them could also be disrupted by changes in β cell clustering within islets. I think that these additional coupling mechanisms deserve to be mentioned in the introduction and be briefly touched upon in the discussion. In fact, since Cx36 expression/distribution pattern does not seem to be affected in this model, these other modes of communication may account for at least a part of the explanation for all possible putative mechanisms, not only diffusible factors. Some of the additional useful references in this regard are:

Eberhard and Lammert, Curr Opin Gen Develop 2009.

Skelin et al., Islets 2017.

Benninger et al., J Physiol 2011.

Benninger and Hodson, Diabetes 2018.

Rutter and Hodson, Mol Endo 2014.

Benninger and Piston, Trend Endocrinol Metab 2014.

9. In Results, the authors support their choice of intravital islet imaging by the fact that during isolation, islets of Robo betaKO mice dissociate, rendering them unsuitable for in vitro analyses of whole-islet Ca oscillations. However, the acute tissue slice method (Speier and Rupnik, Eur J Physiol 2005) has been developed to enable in vitro studies in genetic models with (even more) strongly disrupted architecture. This method was successfully coupled with multicellular confocal Ca imaging (Stozer et al., PLoS One 2013) and is suitable for highest spatial and temporal resolution Ca recordings and I believe that it should be at least briefly mentioned as an option for future studies of Ca imaging in this genetic model. Another experimental solution for cases like this are islets transplanted into the anterior chamber of the eye (Speier et al., Nat Protocol 2008). Both the tissue slice and ACE transplantation method are closer to the in vivo situation in terms of architecture than isolated islets. Of course, in vivo recordings are a step even further, but may not be required in all cases in the future. Moreover, the limits of temporal resolution in an in vivo setup (0.2 Hz in your case) may not allow for analyses of more detailed aspects of intra-islet Ca signalling, such as speed waves, pacemaker location, network analyses etc.

10. To account for the possibility of intrinsic effects of Robo KO on the ability of β cells to respond with Ca oscillations, the authors resorted to imaging Ca oscillations in dissociated β cells. It should be pointed out that this process may per se influence the ability of any β cell to produce Ca oscillations and the nature of the oscillations in isolated and coupled cells may differ importantly (fast vs. slow). Whilst this experiment is important, the fact that some islets or some parts of islets in KO mice showed well synchronized activity in vivo is a much stronger indicator that KO β cells are not intrinsically unable to respond with normal Ca oscillations, given that the Robo gene is truly knocked-out (evidence for this is provided in the Discussion)! My suggestion would be to put the in vitro experiments first, and then the in vivo observations as the more convincing second argument/line of evidence.

Also, a short discussion on the nature of observed Ca oscillations is warranted. Are the in vivo observed oscillations fast or slow or mixed? Sometimes, fast oscillations are superimposed on slower ones. The fast may have periods as low as 2-3 seconds. Therefore, some may have been missed. Therefore, a recording speed of 2 frames per 10 seconds may not be sufficient in all cases.

11. Some of the presented traces seem to provide enough spatiotemporal detail to quantify cells that responded first during a given Ca oscillation and the ones that followed. Also, the speed of Ca »waves« could be assessed, quantified, and compared between CTRLs and KOs. Please comment briefly.

12. Functional connectivity patterns would also be interesting to assess to provide an additional rough idea on the degree of coupling and changes to it (Stozer et al., PLoS Comput Biol 2013, Hodson et al., JCI 2013). Are your data able to support such analyses, at least in the future? Since you have calculated R values, this seems to be a realistic option.

13. The authors should examine the innervation of the ROBO KO islets and make sure that innervation is normal. An aberrant innervation can explain the Ca^2+^ phenotype of ROBO KO islets and it would argue against the central claim that defects in homotypic interactions are responsible for the reduced synchronicity (and not an external factor such as aberrant innervation).

14. In terms of vascularization of ROBO KO islets, a functional test should be performed in order to check for proper vascular delivery of glucose using the 2-NBDG uptake-assay, as described previously by Jacob et al., (PMID: 31914664).

15. The title is misleading since 36% of Robo betaKO are fully synchronized. Additionally, synchronization between the β cells in a normal islet is not a universal and required feature of β cell activity. At physiological glucose concentrations and at the beginning of the recording, the networks of β cells in an islet are preferentially segregated and they only integrate, leading to fully synchronized activity, only at unphysiologically high glucose and prolonged stimulations (Stožer et al., 2013 PLOS One, Markovic et al., 2015, Sci Rep, Stožer et al., 2019, Front Physiol).

16. It is a pity the KCl traces are not shown. The question is, why should KCl stimulus trigger an improved response and glucose not? All steps distal to membrane depolarization should actually be the same.

17. Statistics presented are slightly redundant (text, figure, figure legend) and inconsistent. For example, p-value significance is different in the text referring to Figure 1D, as it is in the figure 1D.

---

## [Author Response]

Although of interest, the manuscript is quite preliminary in many respects, and new work is required to make a compelling case for publication. As detailed below, essential revisions will therefore involve: assessment of the impact of Robo1/2 deletion on Connexin distribution; the impact of deletion on gap junction coupling using dyes; and assessment of the possibility that heterogeneity in calcium wave conduction between islets reflects differences in the extent of disruption of islet architecture (degree of inter-mingling of endocrine cell types). More careful and extensive description of the literature, particularly around intercellular coupling, network behaviour etc is also required (and see also further comments at the foot of this letter)1. The authors infer that preserved calcium waves in a proportion of Robo KO islets likely reflects preserved architecture. This needs to be checked post facto by appropriate staining of the different cell types

We certainly agree with the reviewers and thank them for this important comment. As discussed in our correspondence with the Editors, Robo β-KO islets are difficult to study in vitro because they fall apart during isolation. We have attempted to visualize coordinated calcium oscillations in acute pancreatic slices in collaboration with the labs of Matt Merrins (Wisconsin), Richard Benninger (Colorado), and Joana Almaca (Miami). Unfortunately, using the slice approach, none of our collaborators observe slow β cell calcium oscillations we see in our intravital experiments, making it difficult to compare between approaches. To the best of our knowledge, the only report of oscillatory activity observed in pancreatic slices is from a bioRxiv preprint by the Stozer and Rupnik labs (DOI: 10.1101/2020.03.11.986893v1), and in this case the calcium waves are fast oscillations (and examined without post-imaging immunofluorescence). We agree that post-facto staining of Robo β-KO would add a strong support of our argument that loss of synchronous Ca^2+^ oscillation in our model is directly related to the changes in islet architecture, and in a normal situation we would have asked Stozer and Rupnik to collaborate with us. However this is not an option here as both are Reviewers on our manuscript. As requested by the Editors in our correspondence regarding this point, we now acknowledge these caveats and suggest that future experiments will be needed to address this in lines 354-364 of the revised PDF.

2. On the same theme, some ROBO islets show synchronous intra-islet regions. Moreover, some ROBO islets such as the one shown in Movie 6 are indistinguishable from WT in terms of Ca^2+^ oscillations. Are the regions/islets, which show synchronous oscillations indeed the ones with preserved architecture i.e. normal homotypic interactions of the β-cells? This can be examined by Ca^2+^ imaging followed by IHC analysis of the islets (as in Figure 1) – provided that the same islet can be monitored using imaging and then processed for IHC.

We attempted to employ the acute pancreas tissue slice method in order to stain islets post hoc for architectural analysis after calcium imaging. Unfortunately, for the reasons mentioned above, we were unable to employ this method to Robo βKO islets. In response, we now discuss this limitation details in the text and suggest future experiments in which this may be possible in lines 354-364 (see also response to Reviewers’ comment 1).

3. Acquisition speeds are very low (max 0.2 Hz) so that – as the authors accept – only very low speed Ca^2+^ waves can be detected. I am not full persuaded that imaging throughout the islet in 3D is justified. This limits the conclusions that can be drawn from the study, and comparisons to many recent reports (see 3/) where acquisition times >1 Hz are used. Surprisingly, recent and earlier reports from the laboratories of Stozer and Rupnik (see below), Hodson, JCI, 2013; Rutter, Cell Metab, 2016; Ninov, Nat Metab, 2019 on network dynamics and hub/follower behaviour are missing. The authors are rightly cautious about discussing the possibility that Robo deletion impacts these behaviours given the limitations of the data acquisition (low speed), but this work still needs to be properly cited and put into context.Moreover, if a higher sampling reduces the measured coordination even further, than this coordination is likely not physiologically relevant. The images as collected a 512x512, but smoothed 5x5 for analysis. Would it not be better to collect 100x100 images directly and image 25x faster to check the physiological significance of coordination of oscillations? Discussion and comparisons required.

We thank the reviewers for this comment. To address the point regarding the speed of acquisition, we have performed an analysis on simulated islets at a range of sampling speeds from 10Hz to 0.1Hz to see if identification of intrinsic frequency or functional populations is altered at slower imaging speed (see Author response image 1). This analysis showed that intrinsic frequencies identified at 10Hz were virtually unchanged at 0.1Hz, and that only a modest loss in retainment of cells identified at 10Hz occurred at sampling of 0.1Hz, though most were retained at 1Hz and 0.2Hz. Thus, though our imaging speed may be modestly less sensitive than 10Hz, the quality of the intrinsic frequency measured remains similar. Further, we measured functional connections between cells at this range of sampling speeds and saw that the quality of these measurements did not change dramatically from 10Hz to 0.1Hz and that retainment of functional connection measurements was only modestly decreased from 10Hz to 0.1Hz. We have added detailed discussion on the limits of our imaging speed with reference to this simulation analysis in lines 416-431. In addition, we performed a set of intravital experiments in which we imaged a single islet plane at 1Hz. We observed the same patterns of oscillations compared to islets imaged at slower imaging speeds (now shown in Figure 4—figure supplement 2A). Thus, we would like to argue that our slower imaging speeds accurately measures general qualities of the islet that are visible at 10Hz with only a modest loss in sensitivity. To strengthen this argument further, we performed a network analysis as was done by Stozer et al., (PLoS Comp. Biol., 2013) on calcium videos imaged from 0.1-1Hz to compare these network properties between Robo β-KO and control islets. These analyses are now included in the Results section as two completely new figure (Figure 5, Figure 4—figure supplement 2), and discussed in lines 206-242 in Results and lines 337-354 in Discussion. We have also included detailed reference to hub cells in the introduction in lines 81-84, and a discussion of the results in lines 341-347.

**Author response image 1. sa2fig1:** 

4. Limited expression of GCamP6 across the islet e.g. Supp. Figure 2 poses a problem in terms of accurately reporting Ca^2+^ wave transduction. Fuller description of this limitation and its impact are needed

This point in now addressed in detail in lines 432-443 of the discussion.

5.The assessment of Cx36 expression is Figure 6 is not wholly compelling: is the protein actually localised at the plasma membrane or some other (intracellular?) site. Higher resolution light or electron microscopy are needed. The authors provide a hypothesis that preferential homotypic contact allows for a necessary amount of gap junctions to form. Observed reduction of homotypic contacts between the β cells is not associated with the reduction of Cx36 expression, and it is therefore not clear what is meant by the missing necessary amount of gap junctions? Furthermore, colocalization of Cx36 should not be considered equal to functional gap junctions, and smaller fraction of homotypic nearest neighbors in Robo betaKO β cells does not necessarily equal limited capacity to electrically couple. The question is how is the junctional conductance distributed within an islet? The published data describes a complex temporal pattern of fluctuations in junctional conductance between the β cell pairs, but overall islet junctional conductance is conserved (Jusup et al., 2020, Biophys J).

We agree with the Reviewers on this important point. Accordingly, we have now performed higher resolution confocal imaging for Cx36 to ensure that localization of gap junctions was indeed between β cells. This new data is now included in updated Figure 7. We have also removed the confusing language mentioned above and toned-down claims that loss of gap junction conductance per se is the cause of disrupted calcium oscillations in vivo.

6. The claim that Gap junctions are properly formed in ROBO KO islets needs substantiation using a functional assay. For example, the authors should apply fluorescence recovery after photobleaching to assess GJ function, as described before by Farnsworth et al. (PMID: 25172942), or other means to monitor GJ permeability.

We agree with the reviewer on this comment. However, as mentioned above, Robo βKO islets are fragile in culture and thus not suited to FRAP in that context, and unfortunately our in vivo imaging platform does not support FRAP at this time. We would argue however that the subset of Robo βKO islets that show high synchronicity in Ca^2+^ oscillations suggests that gap junctions are likely able to form between β cells when Robo is deleted. We have now toned-down language surrounding gap junction conductance and noted in lines 354-364 that FRAP in acute tissue slices would be an important future experiment to test gap junction functionality directly.

7. In the Introduction, the view that mouse/rodent islets are of the classical mantle-core type may be a bit too simplified. I suggest that the authors consider/point out the possibility that at least some of the islets, at least during some normal/pathological developmental stages may display other architectures, of which some may be close to the more complex human architecture. I strongly believe that the study could benefit from this addition, since the authors strongly argue for the importance of structure in determining function. In other words, architectural patterns deviating from the textbook core-mantle paradigm may indicate important adaptations to normal (pregnancy, age) or pathological (hyperglycemia) stimuli. For sure, this is something that adds to the importance of their study. Some of the additional useful references are for example:Dolensek et al., Islets 2015.Steiner et al., Islets 2010.Orci, Lancet 1975.Pfeifer, Struct Biol 2015.Erlandsen, Histochem Cytochem 1976.Kharouta, Diab Res Clin Pract 2009.

We thank the reviewer for bringing up this important point. We have now added detailed review on this to the introduction to discuss the ways in which islet architecture can deviate from the norm in both mice and humans, and how this may be important for islet function in the face of changing energy demands, citing all the above references (now refs. number 11, 24, 1, 2, 3, and 17, respectively; see lines 44-49).

8. In the introduction, gap-junctional coupling is pointed out as the only important synchronizing mechanism affected by homotypical interactions/cell-cell contacts. A number of other structural and functional coupling mechanism have been proposed to operate in islets and practically all of them could also be disrupted by changes in β cell clustering within islets. I think that these additional coupling mechanisms deserve to be mentioned in the introduction and be briefly touched upon in the discussion. In fact, since Cx36 expression/distribution pattern does not seem to be affected in this model, these other modes of communication may account for at least a part of the explanation for all possible putative mechanisms, not only diffusible factors. Some of the additional useful references in this regard are:Eberhard and Lammert, Curr Opin Gen Develop 2009.Skelin et al., Islets 2017.Benninger et al., J Physiol 2011.Benninger and Hodson, Diabetes 2018.Rutter and Hodson, Mol Endo 2014.Benninger and Piston, Trend Endocrinol Metab 2014.

We have now added a broader discussion of these factors and their possible role in disrupted calcium oscillations in the Discussion in lines 365-392, which we feel is a better place to discuss this than the Introduction. We included the above reference in this discussion (now refs. number 70, 37, 80, 73, 74, and 30, respectively). We have also added Figure 6—figure supplement 1 (referred to in text in lines 245-251) which examines and discusses innervation in Robo βKO and control islets. Additionally, we have performed a whole new functional assay (2-NBDG analyses in intravital imaging) to assess vascular perfusion within the islet in vivo as well which has been added as new panels to Figure 6 and in text in lines 255-273.

9. In Results, the authors support their choice of intravital islet imaging by the fact that during isolation, islets of Robo betaKO mice dissociate, rendering them unsuitable for in vitro analyses of whole-islet Ca oscillations. However, the acute tissue slice method (Speier and Rupnik, Eur J Physiol 2005) has been developed to enable in vitro studies in genetic models with (even more) strongly disrupted architecture. This method was successfully coupled with multicellular confocal Ca imaging (Stozer et al., PLoS One 2013) and is suitable for highest spatial and temporal resolution Ca recordings and I believe that it should be at least briefly mentioned as an option for future studies of Ca imaging in this genetic model. Another experimental solution for cases like this are islets transplanted into the anterior chamber of the eye (Speier et al., Nat Protocol 2008). Both the tissue slice and ACE transplantation method are closer to the in vivo situation in terms of architecture than isolated islets. Of course, in vivo recordings are a step even further, but may not be required in all cases in the future. Moreover, the limits of temporal resolution in an in vivo setup (0.2 Hz in your case) may not allow for analyses of more detailed aspects of intra-islet Ca signalling, such as speed waves, pacemaker location, network analyses etc.

We have now added a discussion on the use of acute pancreas slices for future experiments in lines 354-364. Further, we have performed analysis of the effect of slower imaging speed on ability to quantify calcium dynamics using simulated islets and found that our imaging speeds only have a modest loss in sensitivity compared to imaging at speeds as fast as 10Hz (see Author response image 1). A detailed discussion of these simulations has been added to lines 426-431.

10. To account for the possibility of intrinsic effects of Robo KO on the ability of β cells to respond with Ca oscillations, the authors resorted to imaging Ca oscillations in dissociated β cells. It should be pointed out that this process may per se influence the ability of any β cell to produce Ca oscillations and the nature of the oscillations in isolated and coupled cells may differ importantly (fast vs. slow). Whilst this experiment is important, the fact that some islets or some parts of islets in KO mice showed well synchronized activity in vivo is a much stronger indicator that KO β cells are not intrinsically unable to respond with normal Ca oscillations, given that the Robo gene is truly knocked-out (evidence for this is provided in the Discussion)! My suggestion would be to put the in vitro experiments first, and then the in vivo observations as the more convincing second argument/line of evidence.Also, a short discussion on the nature of observed Ca oscillations is warranted. Are the in vivo observed oscillations fast or slow or mixed? Sometimes, fast oscillations are superimposed on slower ones. The fast may have periods as low as 2-3 seconds. Therefore, some may have been missed. Therefore, a recording speed of 2 frames per 10 seconds may not be sufficient in all cases.

We thank the reviewers for this important suggestion. Accordingly, we have now changed the order such that the in vitro experiments precede the in vivo experiments. We have also now added a detailed discussion on the nature of the oscillations we have observed in vivo (lines 393-415), and in Supplementary file 2.

11. Some of the presented traces seem to provide enough spatiotemporal detail to quantify cells that responded first during a given Ca oscillation and the ones that followed. Also, the speed of Ca »waves« could be assessed, quantified, and compared between CTRLs and KOs. Please comment briefly.

We thank the reviewers for this comment. We have now performed new network and wave speed/phase lag analyses. These new data are presented in Figures 5 and Figure 4 —figure supplement 2 and discussed in the text in lines 197-199 and in 206-242.

12. Functional connectivity patterns would also be interesting to assess to provide an additional rough idea on the degree of coupling and changes to it (Stozer et al., PLoS Comput Biol 2013, Hodson et al., JCI 2013). Are your data able to support such analyses, at least in the future? Since you have calculated R values, this seems to be a realistic option.

We have now added completely new network analysis, done as described in Stozer et al., (PLoS Comp. Biol., 2013), on calcium videos imaged at frequencies ranging from 0.1Hz to 1Hz in order to compare these network properties between Robo βKO and controls. These are now included with the appropriate citation in the Results section in lines 206-242 and in the newly added Figure 5 and Figure 4—figure supplement 2.

13. The authors should examine the innervation of the ROBO KO islets and make sure that innervation is normal. An aberrant innervation can explain the Ca^2+^ phenotype of ROBO KO islets and it would argue against the central claim that defects in homotypic interactions are responsible for the reduced synchronicity (and not an external factor such as aberrant innervation).

We have now included new experimental assessment of innervation in Robo βKO and control islets through immunohistochemistry with anti-TUBB3 (a broad marker of neurons), showing normal innervation in Robo βKO islets. The results of these new experiments are now included in Figure 6—figure supplement 1, in lines 245-251 of the Results, and in lines 318-322 in the Discussion.

14. In terms of vascularization of ROBO KO islets, a functional test should be performed in order to check for proper vascular delivery of glucose using the 2-NBDG uptake-assay, as described previously by Jacob et al., (PMID: 31914664).

We thank the reviewers for bringing up this point. We have now performed new experiments to address it. Briefly, intravital imaging with tail vein IV injection of 2-NBDG was performed to measure timing and scope of glucose delivery within the islets. These new data are now included in Figure 6 and discussed in lines 255-273 of the Results section and lines 322-326 in the Discussion.

15. The title is misleading since 36% of Robo betaKO are fully synchronized. Additionally, synchronization between the β cells in a normal islet is not a universal and required feature of β cell activity. At physiological glucose concentrations and at the beginning of the recording, the networks of β cells in an islet are preferentially segregated and they only integrate, leading to fully synchronized activity, only at unphysiologically high glucose and prolonged stimulations (Stožer et al., 2013 PLOS One, Markovic et al., 2015, Sci Rep, Stožer et al., 2019, Front Physiol).

We thank the reviewers for this comment. Accordingly, we have now changed the title to “Reduced synchroneity of intra-islet Ca^2+^ oscillations in vivo in Robo-deficient β cells” to reflect this aspect of calcium dynamics.

16. It is a pity the KCl traces are not shown. The question is, why should KCl stimulus trigger an improved response and glucose not? All steps distal to membrane depolarization should actually be the same.

We agree with the Reviewers on this point. Accordingly, we have now added KCl traces to Figure 2. Additionally, upon re-evaluation of our data, we saw that one KO mouse in particular had quite a disproportionate KCL response compared to its matched control while the other 3 out of 4 pairs showed no difference. We thus repeated the experiment with an additional pair of mice to increase our n and get a better picture of the real biological effect of Robo KO on KCL AUC. After adding this pair, KCL AUC is no longer significantly different between pooled control and KO β cells. This is more in line with what we see in each individual experiment, namely that in 4 out of 5 mouse pairs that we assayed, there was no significant difference observed in the KCL response.

17. Statistics presented are slightly redundant (text, figure, figure legend) and inconsistent. For example, p-value significance is different in the text referring to Figure 1D, as it is in the figure 1D.

We have now eliminated the redundancy in the statistics reporting by moving it exclusively into figure legends. We now also more rigorously define our data analysis methods with the appropriate statistical tests in the cases were this was not done. We further added a detailed statistical analysis methods section to define our methods more concretely in lines 685-690.